# Boltzmann Semantic Score: A Semantic Metric for Evaluating Large Vision Models Using Large Language Models

**Ali Khajegili Mirabadi, Katherine Rich, Hossein Farahani, & Ali Bashashati**
School of Biomedical Engineering & Department of Pathology and Laboratory Medicine
The University of British Columbia
`{ali.mirabadi,ali.bashashati}@ubc.ca`

## Abstract

Do Large Vision Models (LVMs) extract medically and semantically relevant features similar to those identified by human experts? Currently, only biased, qualitative approaches with limited, small-scale expert evaluations are available to answer this question. In this study, we propose the Boltzmann Semantic Score (BSS), a novel method inspired by state space modeling, to evaluate the encoding space of LVMs from medical images using the encoding space of Large Language Models (LLMs) from medical reports. Through extensive experimentation on 32 datasets from The Cancer Genome Atlas collection using five state-of-the-art LLMs, we first establish a baseline of LLMs' performance in digital pathology and show that LLMs' encoding can be linked to patient outcomes. Then, we compared seven LVMs with BSS and showed that LVMs suffer from poor semantic capability when compared with encoded expert knowledge from pathology reports. We also found statistically significant correlations between BSS (as a measure of structural similarity) and performance in two downstream tasks: information retrieval and survival prediction tasks. Our study also investigates the consensus among LLMs in evaluating LVMs using BSS, indicating that LLMs generally reach substantial consensus in rating LVMs, with some variation dependant on the cancer type. We believe the BSS metric proposed here holds significant potential for application in other domains with similar contexts. Data and code can be found in `https://github.com/AIMLab-UBC/Boltzmann`

## 1 Introduction

To even a casual observer, Large Vision Models (LVMs), and Large Language Models (LLMs) seem poised to revolutionize medicine. However, despite their rapid adoption in other domains, they have yet to make a substantial impact within the medical field (Omiye et al., 2024). Currently, they are notably absent from any direct clinical applications. This is due to a variety of distinct challenges inherent to medical data: privacy, lack of large and diverse datasets, ethical considerations, licensing, and the inherent complexity of the domain (Bouderhem, 2024; Jiang et al., 2021). This complexity makes interpreting the results of large models incredibly challenging, with researchers often resorting to qualitative assessment of the model's response to specific and limited inputs.

The standard method used in research studies is to recruit medical experts to examine the model's attention maps or activation layers as a way to determine if they are capturing clinically relevant information. However, this approach has inherent variation due to the limited number of medical experts, and inter-/intra-observer variability. When using these approaches, evaluations can also be biased as studies are often limited to small sample sizes. Variability can also originate from the medical expert's training, specialty, and experience. Additionally, the diagnostic guidelines for various diseases that medical experts rely on are constantly being updated (Alonso-Coello et al., 2011). These changes can impact the relevancy of older models that were trained on labels produced using older guidelines, and evaluated using those same guidelines.

In standard medical practice, imaging data is typically analyzed by a single specialist, except in rare cases that require a collective assessment. In the field of pathology, experts examine the tumor tissue under the microscope or their corresponding images and document their observations in text-based

pathology reports.These pathology reports serve as concise summaries of the most significant findings. Clinicians routinely consult these reports to review a patient's history or diagnosis, especially when investigating rare cases, where one might be interested in identifying historical patients with similar profiles to inform care for the patient in hand(Borowsky & et al., 2020; Evans & et al., 2022; Farooq & et al., 2021).

With the recent rise in popularity of LVMs in digital pathology, an important question arises: do LVMs extract medically and semantically relevant features similar to those identified by human experts? This question remains largely unanswered. However, the rapid advancement of large language models (LLMs) has made it feasible to represent long and complex contexts, such as detailed pathology reports as illustrated in Figure 1 using UMAPs (McInnes et al., 2018). This opens up the possibility of creating a large database of medical reports, encoding them with LLMs, and developing a comprehensive resource of encoded medical information. By leveraging this re-

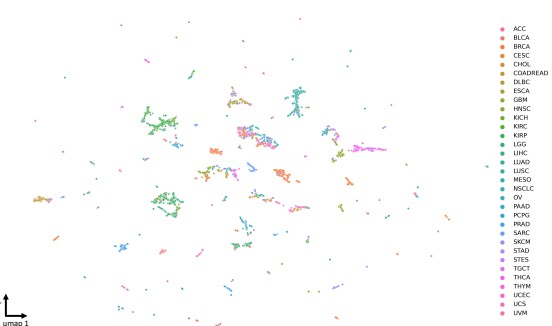

Figure 1: The UMAP plot of the encodings of Command-R, an open-source LLM, from roughly 9,500 patients' pathology reports over 32 different datasets in The Cancer Genome Atlas. Clusters indicate clear differentiation between cancer types in the encoding space of Command-R, showing the representation capability of LLMs.

source, we can address the question posed earlier. Evaluating LVMs on this extensive set can mitigate biases inherent in smaller sample sizes and incorporate input from multiple observers (pathologists), thereby reducing observer variability.

To this end, our primary contribution is to propose a systematic methodology, independent of the choice of LLM or LVM, to evaluate LVMs from a medically semantic perspective. Our methodology leverages LLMs and a large and collective database of medical reports across more than 30 cancer types that represent more than 9,500 patients. We also establish a baseline of LLMs' performance in two large-scale digital pathology tasks.

## 2 BACKGROUND & RELATED WORKS

The term *foundation model* was first coined by Bommasani et al (Bommasani et al., 2021) to describe models trained on huge and broad datasets that can be easily adapted to a variety of downstream tasks. Since then, multiple foundational models have been introduced for a variety of domains. For text, GPT-3 (Brown et al., 2020), Jamba (Lieber et al., 2024), Gemma-7b (Team et al., 2024), Llama3-8b (Touvron et al., 2023; AI@Meta, 2024), Bio-Llama3-8b (Ankit Pal, 2024), and Command-R (CohereForAI, 2024) are a few examples to name. Within computer vision and vision-language, general examples include CLIP (Radford et al., 2021) and VL-BERT (Su et al., 2019). However, in recent years, beyond generic models such as ViT (Dosovitskiy et al., 2020) and Swin (Liu et al., 2021), a wave of domain-specific vision LVMs have been introduced. Within the medical field and more specifically in the histopathology imaging space, examples of LVMs include CTransPath (Wang et al., 2022), PLIP (Huang et al., 2023), Phikon (Filiot et al., 2023), UNI (Chen et al., 2023), and Virchow (Vorontsov et al., 2023).

To evaluate the performance of these models, they are often scored on an ensemble of downstream medical tasks. Common examples of these tasks include tumor grading, tumor subtype classification, image segmentation, and disease classification. While the performance of a model on these tasks can give us an idea of how it performs in comparison to other machine learning models, it falls short of being conclusive evidence that the model will achieve similar performance in a clinical setting. Thus, in order to offer further insight into how the model is 'thinking', researchers will use qualitative techniques such as attention-mapping (Bahdanau et al., 2014) or GradCAM (Selvaraju et al., 2017) to highlight which sections of the input data the model finds most important for the relevant task. From there, one can attempt to draw direct comparisons to how a clinician will parse the same data, which is subject to bias.

In the context of medicine, where reports are typically written according to specific clinical guidelines, variability persists due to diverse writing styles, training backgrounds, professional experiences, and

institution-specific protocols. However, LLMs, owing to their exposure to diverse training text data, are adept at tackling these variabilities (Singhal et al., 2023; Ankit Pal, 2024; Luo et al., 2022; Achiam et al., 2023; Lieber et al., 2024). As a result, LLMs demonstrate the potential to encode medical information effectively, providing nuanced representations that encapsulate expert knowledge. Consequently, the encoded medical report representations generated by LLMs serve as a valuable source of medical assessments for each paired image.

Accordingly, we propose the Boltzmann Semantic Score (BSS) where we utilize LLMs to encode medical reports accompanying digital images and then pair the feature spaces of LLM and LVM to measure the similarity of visual representations to the textual representations. This similarity indicates a patient-to-patient relationship, where we expect the same relationship between patients captured by experts (through medical reports) to be captured by LVMs.

Here, we argue that BSS is a measure of the structural similarity between the visual embedding space and the text embedding space, where the text embedding space represents medical semantics as the reference. It can be interpreted as a measure of semantic similarity for LVMs within the broader medical domain.

## 3 METHOD

This section begins by introducing the intuition behind our work, followed by the problem formulation, and then expands the theory behind the Boltzmann Semantic Score.

### 3.1 INTUITION

We start with an example to illustrate the intuition behind our method: consider only three cases—$A$, $B$, and $C$—each comprising paired text and image data in our dataset. Expert annotations through an LLM indicate that case $A$ is semantically closer to $B$ than to $C$, reflecting a stronger clinical correspondence between $A$ and $B$, with a weaker but non-negligible similarity between $A$ and $C$.

An ideal LVM should reflect these relationships by embedding $A$ closer to $B$ than to $C$ within its latent space. When this alignment is achieved, the semantic similarity for $A$ ($\mathcal{B}_A$) reaches its maximum value ($\mathcal{B}_A = 1$), signifying strong semantic agreement with expert-provided annotations. Conversely, if the LVM incorrectly positions $A$ closer to $C$ than to $B$, the semantic score for $A$ should decrease. Importantly, due to inherent baseline similarities among cases, the score should not sharply drop to zero but rather be proportional to their energy distribution. With this mechanism, by averaging the semantic score values across the entire dataset, we obtain an overall score that quantifies the LVM's alignment with the reference LLM. Using an LLM as the fixed reference, the semantic score obtained allows us to compare different LVMs, highlighting which model best aligns its latent space with the semantic structure of the LLM and most effectively preserves semantic information.

### 3.2 PROBLEM FORMULATION

For a paired set of $n$ texts and images in a dataset, we can represent the set of the text data as $\mathcal{T} = \{\mathcal{T}_t \mid t \text{ is an integer}, 1 \leq t \leq n\}$, and the set of the image data as $\mathcal{I} = \{\mathcal{I}_r \mid r \text{ is an integer}, 1 \leq r \leq n\}$, where $\mathcal{T}_t$ and $\mathcal{I}_r$ are the matching pairs for any given $t = r$. With this definition, let's denote any given LLM as $\Lambda(\mathcal{T}_t)$ representing a function of input text $\mathcal{T}_t$, and any given LVM as $\Phi(\mathcal{I}_r)$ representing a function of input image $\mathcal{I}_r$. Therefore, $\Lambda$ and $\Phi$ are defined as,

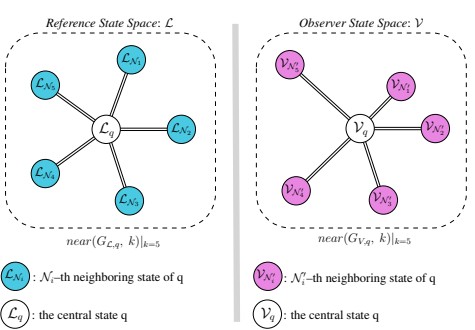

Figure 2: The state space representation of the k–nearest neighbors in (left) text space ($\mathcal{L}$) and in (right) the vision space ($\mathcal{V}$). The *observer* is estimating states that matches to *reference*.

$$\Lambda : \mathcal{T} \longrightarrow \mathcal{L} \subseteq \mathbb{R}^{d_1 \times 1}$$
$$\Phi : \mathcal{I} \longrightarrow \mathcal{V} \subseteq \mathbb{R}^{d_2 \times 1}$$

(1)

where $\mathcal{L}$ and $\mathcal{V}$ are the set of dense representations of the input text and image tensors, respectively. Similar to $\mathcal{T}$, $\mathcal{L} = \{\mathcal{L}_t \mid t \text{ is an integer}, 1 \leq t \leq n\}$; and similar to $\mathcal{I}$, $\mathcal{V} =$

$\{\mathcal{V}_r \mid r \text{ is an integer}, \ 1 \leq r \leq n\}$. $\mathbb{R}^{d_1 \times 1}$ and $\mathbb{R}^{d_2 \times 1}$ are the set of $d_1-$ and $d_2-$dimensional vector space, respectively. And, $d_1$ and $d_2$ are the number of features of the dense representation of $\Lambda$ and $\Phi$, respectively. Note that most often $d_1 \neq d_2$.

The end goal here is to find a function $\mathcal{F}$ such that assigns a relative semantic score to $\Phi$ using $\Lambda$ on the paired dataset premised upon their dense representations. Given an instance of query $q$, we expect $\mathcal{B}_q = \mathcal{F}(\mathcal{V}_q ; \mathcal{L})$, where $\mathcal{B}_q$ is the desired score measuring the semantic worth of information extracted from the query image using $\Phi$ compared to the query medical text written by human experts embedded by $\Lambda$.

### 3.3 BOLTZMANN SEMANTIC SCORE

To come up with score function $\mathcal{F}$, for the query $q$, we define the set of the k-nearest neighbors of $\mathcal{L}_q$ as $near(\mathcal{L}_q, k) = \{\mathcal{L}_{\mathcal{N}_i} \mid 1 \leq \mathcal{N}_i \leq k\}$, and the set of the k–nearest neighbors of $\mathcal{V}_q$ as $near(\mathcal{V}_q, k) = \{\mathcal{V}_{\mathcal{N}_i'} \mid 1 \leq \mathcal{N}_i' \leq k\}$. Hereafter, for the sake of simplicity in our notation, we refer to $\mathcal{L}_{\mathcal{N}_i}$ as $\mathcal{L}_i$ (the $i$–th nearest neighbor to $\mathcal{L}_q$ in $\mathcal{L}$), and to $\mathcal{V}_{\mathcal{N}_i'}$ as $\mathcal{V}_j$ (the $j$–th nearest neighbor to $\mathcal{V}_q$ in $\mathcal{V}$).

Now, we can define two Star graphs corresponding to each query node in the text and vision spaces: *Reference* and *Observer*. For the text, which is the *reference*, $G_{\mathcal{L},q}$ is defined with the central node of $\mathcal{L}_q$ where the leaves are members of $near(\mathcal{L}_q, k)$. Similarly, the vision space, which is the *observer*, $G_{\mathcal{V},q}$ is defined with a central node of $\mathcal{V}_q$ and leaves as members of $near(\mathcal{V}_q, k)$. It is worth mentioning that the k–neighbors of query $q$ in the text and vision spaces are not essentially equal as they are calculated using different text and vision encoders. Example graphs are illustrated in Figure 2.

Here, $G_{\mathcal{L},q}$ is a state graph representing the relationship between the text encodings written by human experts after visually inspecting the corresponding images. The individual nodes can be viewed as a quantum state that ties in medical semantics; thus, it is expected that $\Lambda$ can properly encode them. That is to say, $G_{\mathcal{L},q}$ represents the relationship between states sharing common information. Thus, if two states/nodes are in close proximity in the state space compared to other states/nodes, the semantic similarity between the two states/nodes is more meaningful than the others. Here, $G_{\mathcal{L},q}$ represents this for the k–nearest states/nodes in the text state space as the *reference*.

On the other hand, $G_{\mathcal{V},q}$ is a state graph representing the relationship between the encoding of images produced by $\Phi$. If we expect an LVM to perform equivalently to human experts, it should ideally match $G_{\mathcal{L},q}$. In other words, the same relationship between states/nodes captured in $G_{\mathcal{L},q}$ should be captured by $\Phi$ in $G_{\mathcal{V},q}$, meaning that the *observer* aligns with the *reference*.

In statistical mechanics, the quantity representing the variety of states associated with a specific energy level is termed the degree of degeneracy, often referred to simply as the level's degeneracy (Shankar, 1994). Using this concept, we want to compare $G_{\mathcal{V},q}$ to $G_{\mathcal{L},q}$ as well as measure their similarity. When comparing these state spaces, if there is a matching state, we consider that as one degree of degeneracy. However, if there is no matching state (a state is in $G_{\mathcal{V},q}$ that is not present in $G_{\mathcal{L},q}$) we need to have two degrees of degeneracy to model that. To clarify, let's say $\mathcal{L}_j$ is a non-matching state in $G_{\mathcal{L},q}$, corresponding with $\mathcal{V}_j$, where $\mathcal{L}_i$ is the true matching state in $G_{\mathcal{L},q}$. Thus, there should be a mechanism to incorporate the randomness between the states $\mathcal{L}_i$ and

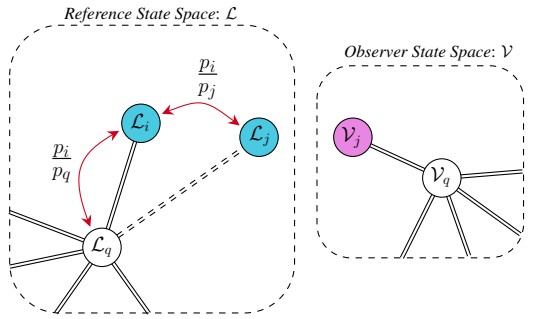

$\mathcal{L}_i$ : the reference matching state

$\mathcal{L}_j$ : the corresponding non-matching estimated state by $\Phi$ in $\mathcal{L}$

$\mathcal{V}_j$ : the estimated state $j$ by $\Phi$ in $G_{\mathcal{V},q}$

Figure 3: The Second-order Boltzmann Factor elucidates the resemblance between states $i$ and $q$ by incorporating the stochastic displacement of states $j$ from $i$. Essentially, $\frac{p_i}{p_j}$ serves as a metric indicating the degree of similarity in energy between these states, thereby causing the model $\Phi$ to displace them accordingly.

$\mathcal{L}_j$ when measuring the total similarity of the two state spaces, as shown in Figure 3.

Having introduced the concept above, we model the distribution of the states with Boltzmann Distribution in the *reference* state space $G_{\mathcal{L},q}$, which dates back to the famous Boltzmann's paper in 1877 (Boltzmann, 1877) with a simplified derivation introduced in (Müller, 2014). This expresses that fractions of energy are distributed over the discrete energy levels of an assembly of states. Therefore, for the central state/node $\mathcal{L}_q$ and a given neighbor $\mathcal{L}_i$ in $G_{\mathcal{L},q}$, the Boltzmann Factor indicates the similarity of the states (in terms of energy) within the state space. As the Boltzmann factor is solely dependent on the energy difference between states, we can formulate and adopt it for our problem as follows,

$$\text{Boltzmann Factor:} \quad \frac{p_i}{p_q} = \exp\left(-\frac{\Delta\mathcal{E}}{kT}\right) = \exp\left(-\frac{\|\mathcal{L}_i - \mathcal{L}_q\|_2}{\sqrt{d_1}}\right) \tag{2}$$

where $p_i$ is the probability of state/node $\mathcal{L}_i$, and $p_q$ is the probability of state/node $\mathcal{L}_q$. For the energy level, $\Delta\mathcal{E} = \|\mathcal{L}_i - \mathcal{L}_q\|_2$ represents the energy difference of the two states/nodes, and $\sqrt{d_1}$ is a constant equivalent to $kT$ in the original formula, where $d_1$, as previously introduced, is the *reference* space dimensionality.

To clarify this concept of modeling, when $\frac{p_i}{p_q}$ approaches 1, it means the two states have similar energy levels and can appear in place of each other in the state space. For the *reference* state space, which is the textual representation, it means that the texts for two patients are characteristically similar. Conversely, when $\frac{p_i}{p_q}$ approaches 0, it indicates that the two states are not characteristically similar.

However, the Boltzmann Factor in 2 only represents one degree of degeneracy/similarity between two states. Thus, we add the second order of degeneracy for the states that are in $near(\mathcal{V}_q, k)$ but not present in $near(\mathcal{L}_q, k)$. In other words, those states that the *observer* mistakenly estimated that are present in *reference*. Therefore, for a given state $j$ such that $\mathcal{V}_j \in near(\mathcal{V}_q, k)$ but $\mathcal{L}_j \notin near(\mathcal{L}_q, k)$, and the true matching state is $\mathcal{L}_i$ in the *reference*, we define the second order Boltzmann Factor as follows,

$$\text{Second-order Boltzmann Factor:} \quad b_{i;j|q} := \frac{p_i}{p_j} \cdot \frac{p_i}{p_q} \tag{3}$$

where $\frac{p_i}{p_j}$ shows the similarity between the two state $\mathcal{L}_i$ and $\mathcal{L}_j$; and $\frac{p_i}{p_q}$ shows the the degeneracy/similarity between the two state $\mathcal{L}_i$ and $\mathcal{L}_q$. Consequently, $b_{i;j|q}$ includes the degeneracy between $\mathcal{L}_i$ and $\mathcal{L}_j$ when comparing $\mathcal{L}_i$ with $\mathcal{L}_q$.

Now, since only the presence of states and their energy matters for modeling the energy distributions of the states, we ignore the orders of the states between the two state spaces if their presence correctly is estimated by the *observer*. Hence, we define $\mathbb{A}$ as the set of all the matching states in $G_{\mathcal{L},q}$ and $G_{\mathcal{V},q}$ here:

$$\mathbb{A} = \{(i,j) \mid \forall \mathcal{V}_j \in near(\mathcal{V}_q, k) \,\&\, \mathcal{L}_j \in near(\mathcal{L}_q, k), \,\exists i : \mathcal{L}_i \in near(\mathcal{L}_q, k) \,\&\, \mathcal{L}_i = \mathcal{L}_j\} \tag{4}$$

Nevertheless, we define the set of the non-matching states between $G_{\mathcal{L},q}$ and $G_{\mathcal{V},q}$ as $\mathbb{D}$:

$$\mathbb{D} = \{(i,j) \mid \forall \mathcal{V}_j \in near(\mathcal{V}_q, k) \,\&\, \mathcal{L}_j \notin near(\mathcal{L}_q, k), \,\exists i : \mathcal{V}_i \notin near(\mathcal{V}_q, k) \,\&\, \mathcal{L}_i \in near(\mathcal{L}_q, k)\} \tag{5}$$

For any $(i,j) \in \mathbb{A}$, essentially $\mathcal{L}_i = \mathcal{L}_j$, so the second order Boltzmann Factors is $b_{i;j|q} = \frac{p_i}{p_j} \cdot \frac{p_i}{p_q} = 1 \cdot \frac{p_i}{p_q} = \frac{p_i}{p_q}$, which means that there is only degeneracy between states $i$ and $q$. On the other hand, for any $(i,j) \in \mathbb{D}$, we see $b_{i;j|q} = \frac{p_i}{p_j} \cdot \frac{p_i}{p_q}$ which is showing the stochasticity between states $i$ and $j$ can affect the presence of the states $i$ in proximity to $q$. With this concept, we introduce the normalized Boltzmann Semantic Score $\mathcal{B}_q$ for the query $q$ as follows,

$$\mathcal{B}_q = \frac{\sum\limits_{(i,j)\in\mathbb{A}} b_{i;j|q}}{\sum\limits_{(i,j)\in\mathbb{A}\cup\mathbb{D}} b_{i;j|q}} = \frac{\sum\limits_{(i,j)\in\mathbb{A}\cup\mathbb{D}} b_{i;j|q} - \sum\limits_{(i,j)\in\mathbb{D}} b_{i;j|q}}{\sum\limits_{(i,j)\in\mathbb{A}\cup\mathbb{D}} b_{i;j|q}} = 1 - \frac{\sum\limits_{(i,j)\in\mathbb{D}} b_{i;j|q}}{\sum\limits_{(i,j)\in\mathbb{A}\cup\mathbb{D}} b_{i;j|q}} \tag{6}$$

Finally, the average Boltzmann Semantic Score is calculated as,

$$\mathcal{B} = \frac{1}{|\mathcal{L}|} \cdot \sum_{q\in\mathcal{L}} \mathcal{B}_q \tag{7}$$

where $|\mathcal{L}|$ is the cardinality of $\mathcal{L}$, and $\mathcal{B}$ denotes the average Boltzmann Semantic Score on the entire set. With the formulation above, we expect that $\mathcal{B}$ not only represents the semantic capability of $\Phi$ using $\Lambda$, but also presents a predictive capability for downstream tasks that uses visual encodings.

## 4 EXPERIMENTS

### 4.1 DATASET & COMPUTE

The largest publicly available pathology archive of datasets, The Cancer Genome Atlas Project (TCGA)(Weinstein et al., 2013), was utilized in this study. TCGA archives include 32 different cancer datasets and roughly $12,000$ patients across more than $160$ different centers, including matched Whole Slide Images (WSIs) and pathology reports. For pathology reports, we used the cleaned TCGA-Reports dataset (Kefeli & Tatonetti, 2024), which is the largest publicly available dataset of cleaned pathology reports to date (more details in section D in Appendix). All experiments were performed on 14 nodes of Intel Xeon E5-4640 with 503GB RAM.

### 4.2 EVALUATION METRICS

For the evaluation metrics, we followed the standard metrics used in (Chen et al., 2022; Kalra et al., 2020; Wang et al., 2023), including top-1, top-3, and top-5 majority vote accuracy, top-1, top-3, and top-5 F1 Score, as well as top-1, top-3, and top-5 Average Precision (AP@k) for the retrieval tasks. Additionally, we employed the concordance index (C-index) to assess performance on the survival prediction task.

### 4.3 LLM REPRESENTATIONS

Recent studies have highlighted the promise of LLMs in various pathology tasks (Wiest et al., 2024; Lammert et al., 2024; Huang et al., 2024). Building on these insights and the findings from the original work on the TCGA reports on the quality of parsed pathology reports and the high performance of language models (Kefeli & Tatonetti, 2024), we have designed two additional large-scale experiments focused on content-based information retrieval and survival prediction using only textual data.

In this regard, we first encode the original text reports without any perturbation and conduct a retrieval test on them within the database. Next, we perturb the input text reports by removing essential keywords, such as cancer names or subtypes, and then encode the perturbed reports with $\Lambda$ and conduct the same retrieval test. By comparing the results of these two experiments, we can determine whether LLMs are capable of properly encoding the medical text reports, focusing on the ability to capture the overall context, not just specific keywords. Finally, to assess whether the LLM representations contain clinically relevant features, we evaluated the original text representations on a survival prediction task and studied if they can be linked to patient outcomes.

#### 4.3.1 INFORMATION RETRIEVAL

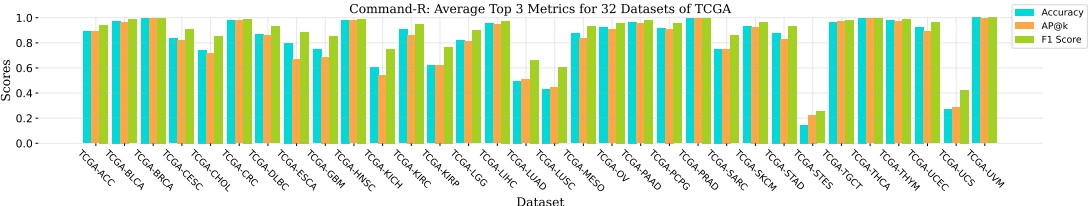

Figure 4: Performance comparison of Command-R, the best-performing LLM, in the organ-independent setting for content-based information retrieval using the original text reports.

In this experiment, we used the LLM representations to conduct a retrieval test. The search was conducted in two settings: *organ-specific* search and *organ-independent* search.

In the *organ-specific* search, we used the cancer type as the query, searching among all cancer types associated with the organ related to the query's cancer type. In the *organ-independent* search, the cancer type was used as the query across all cancer types in the database (The detailed information on the different cancers and their organ allocations is provided in Table 4 and section D in the

Table 1: Information Retrieval with 5 LLMs over 32 Datasets of TCGA

(a) Average Performance of Organ-Specific Search with Original Text Reports

| LLM | Accuracy | | | F1 Score | | | AP@k | | |
|---|---|---|---|---|---|---|---|---|---|
| | top 1 | top 3 | top 5 | top 1 | top 3 | top 5 | top 1 | top 3 | top 5 |
| Jamba | $0.838_{\pm0.17}$ | $0.823_{\pm0.20}$ | $0.811_{\pm0.22}$ | $0.900_{\pm0.12}$ | $0.886_{\pm0.16}$ | $0.874_{\pm0.19}$ | $0.838_{\pm0.17}$ | $0.813_{\pm0.19}$ | $0.795_{\pm0.20}$ |
| Gemma-7b | $0.782_{\pm0.19}$ | $0.750_{\pm0.22}$ | $0.722_{\pm0.25}$ | $0.863_{\pm0.14}$ | $0.835_{\pm0.19}$ | $0.809_{\pm0.22}$ | $0.782_{\pm0.19}$ | $0.738_{\pm0.20}$ | $0.708_{\pm0.21}$ |
| Llama3-8b | $0.852_{\pm0.16}$ | $0.844_{\pm0.17}$ | $0.829_{\pm0.21}$ | $0.911_{\pm0.11}$ | $0.904_{\pm0.12}$ | $0.888_{\pm0.18}$ | $0.852_{\pm0.16}$ | $0.827_{\pm0.17}$ | $0.809_{\pm0.18}$ |
| Bio-Llama3-8b | $0.860_{\pm0.15}$ | $0.850_{\pm0.17}$ | $0.810_{\pm0.20}$ | $0.917_{\pm0.10}$ | $0.909_{\pm0.12}$ | $0.893_{\pm0.16}$ | $0.860_{\pm0.15}$ | $0.839_{\pm0.17}$ | $0.819_{\pm0.19}$ |
| Command-R | $0.871_{\pm0.15}$ | $0.868_{\pm0.17}$ | $0.859_{\pm0.18}$ | $0.923_{\pm0.10}$ | $0.919_{\pm0.12}$ | $0.912_{\pm0.13}$ | $0.871_{\pm0.15}$ | $0.851_{\pm0.17}$ | $0.835_{\pm0.18}$ |

(b) Average Performance of Organ-Specific Search with Perturbed Text Reports

| Cancer | Accuracy | | | F1 Score | | | AP@k | | |
|---|---|---|---|---|---|---|---|---|---|
| | top 1 | top 3 | top 5 | top 1 | top 3 | top 5 | top 1 | top 3 | top 5 |
| Jamba | $0.827_{\pm0.18}$ | $0.811_{\pm0.20}$ | $0.794_{\pm0.23}$ | $0.893_{\pm0.13}$ | $0.875_{\pm0.18}$ | $0.858_{\pm0.21}$ | $0.827_{\pm0.18}$ | $0.800_{\pm0.20}$ | $0.776_{\pm0.22}$ |
| Gemma-7b | $0.762_{\pm0.20}$ | $0.724_{\pm0.21}$ | $0.699_{\pm0.22}$ | $0.848_{\pm0.16}$ | $0.813_{\pm0.18}$ | $0.790_{\pm0.21}$ | $0.762_{\pm0.20}$ | $0.736_{\pm0.21}$ | $0.711_{\pm0.22}$ |
| Llama3-8b | $0.844_{\pm0.17}$ | $0.828_{\pm0.19}$ | $0.813_{\pm0.22}$ | $0.906_{\pm0.12}$ | $0.892_{\pm0.15}$ | $0.876_{\pm0.18}$ | $0.844_{\pm0.17}$ | $0.827_{\pm0.19}$ | $0.809_{\pm0.21}$ |
| Bio-Llama3-8b | $0.852_{\pm0.16}$ | $0.838_{\pm0.18}$ | $0.808_{\pm0.21}$ | $0.911_{\pm0.11}$ | $0.897_{\pm0.13}$ | $0.877_{\pm0.17}$ | $0.852_{\pm0.16}$ | $0.832_{\pm0.18}$ | $0.807_{\pm0.19}$ |
| Command-R | $0.848_{\pm0.17}$ | $0.846_{\pm0.19}$ | $0.815_{\pm0.21}$ | $0.908_{\pm0.11}$ | $0.905_{\pm0.14}$ | $0.883_{\pm0.19}$ | $0.848_{\pm0.17}$ | $0.830_{\pm0.19}$ | $0.804_{\pm0.20}$ |

(c) Average Performance of Organ-Independent Search with Original Text Reports

| LLM | Accuracy | | | F1 Score | | | AP@k | | |
|---|---|---|---|---|---|---|---|---|---|
| | top 1 | top 3 | top 5 | top 1 | top 3 | top 5 | top 1 | top 3 | top 5 |
| Jamba | $0.830_{\pm0.18}$ | $0.782_{\pm0.23}$ | $0.773_{\pm0.24}$ | $0.895_{\pm0.13}$ | $0.855_{\pm0.18}$ | $0.847_{\pm0.20}$ | $0.830_{\pm0.19}$ | $0.788_{\pm0.20}$ | $0.762_{\pm0.21}$ |
| Gemma-7b | $0.731_{\pm0.19}$ | $0.637_{\pm0.23}$ | $0.611_{\pm0.24}$ | $0.829_{\pm0.15}$ | $0.750_{\pm0.21}$ | $0.726_{\pm0.22}$ | $0.731_{\pm0.20}$ | $0.655_{\pm0.20}$ | $0.613_{\pm0.21}$ |
| Llama3-8b | $0.810_{\pm0.20}$ | $0.786_{\pm0.22}$ | $0.766_{\pm0.24}$ | $0.880_{\pm0.15}$ | $0.859_{\pm0.18}$ | $0.842_{\pm0.20}$ | $0.810_{\pm0.21}$ | $0.778_{\pm0.21}$ | $0.755_{\pm0.21}$ |
| Bio-Llama3-8b | $0.815_{\pm0.19}$ | $0.792_{\pm0.21}$ | $0.778_{\pm0.23}$ | $0.884_{\pm0.19}$ | $0.865_{\pm0.21}$ | $0.853_{\pm0.23}$ | $0.815_{\pm0.19}$ | $0.783_{\pm0.21}$ | $0.759_{\pm0.21}$ |
| Command-R | $0.825_{\pm0.20}$ | $0.817_{\pm0.22}$ | $0.801_{\pm0.23}$ | $0.888_{\pm0.15}$ | $0.879_{\pm0.17}$ | $0.866_{\pm0.19}$ | $0.825_{\pm0.21}$ | $0.805_{\pm0.22}$ | $0.787_{\pm0.22}$ |

(d) Average Performance of Organ-Independent Search with Perturbed Text Reports

| LLM | Accuracy | | | F1 Score | | | AP@k | | |
|---|---|---|---|---|---|---|---|---|---|
| | top 1 | top 3 | top 5 | top 1 | top 3 | top 5 | top 1 | top 3 | top 5 |
| Jamba | $0.778_{\pm0.21}$ | $0.752_{\pm0.23}$ | $0.729_{\pm0.25}$ | $0.857_{\pm0.16}$ | $0.834_{\pm0.19}$ | $0.812_{\pm0.23}$ | $0.778_{\pm0.21}$ | $0.748_{\pm0.21}$ | $0.721_{\pm0.22}$ |
| Gemma-7b | $0.638_{\pm0.20}$ | $0.575_{\pm0.22}$ | $0.515_{\pm0.24}$ | $0.760_{\pm0.17}$ | $0.703_{\pm0.20}$ | $0.642_{\pm0.25}$ | $0.638_{\pm0.20}$ | $0.579_{\pm0.20}$ | $0.534_{\pm0.20}$ |
| Llama3-8b | $0.792_{\pm0.20}$ | $0.769_{\pm0.22}$ | $0.736_{\pm0.25}$ | $0.868_{\pm0.15}$ | $0.849_{\pm0.18}$ | $0.820_{\pm0.20}$ | $0.801_{\pm0.21}$ | $0.775_{\pm0.21}$ | $0.755_{\pm0.22}$ |
| Bio-Llama3-8b | $0.791_{\pm0.20}$ | $0.767_{\pm0.22}$ | $0.744_{\pm0.24}$ | $0.867_{\pm0.16}$ | $0.846_{\pm0.19}$ | $0.827_{\pm0.20}$ | $0.790_{\pm0.20}$ | $0.757_{\pm0.21}$ | $0.731_{\pm0.21}$ |
| Command-R | $0.801_{\pm0.21}$ | $0.784_{\pm0.23}$ | $0.768_{\pm0.25}$ | $0.873_{\pm0.15}$ | $0.856_{\pm0.18}$ | $0.844_{\pm0.20}$ | $0.801_{\pm0.21}$ | $0.775_{\pm0.21}$ | $0.755_{\pm0.22}$ |

Appendix, respectively.). For both settings, we utilized both original and perturbed representations to compare the contextual understanding capabilities of LLMs. We report the average results for both organ-specific and organ-independent searches across 32 different datasets in Table 1. To illustrate this further, Figure 4 highlights the performance of the best-performing LLM across various datasets in the organ-independent setting, which is the harder task. The detailed cancer-by-cancer results for all 5 LLMs across different settings are available in section M in the Appendix.

As Table 1a shows, LLMs perform well with Command-R achieving a Top-1 accuracy of $0.871$ and a Top-1 F1 Score of $0.919$ on average across all the cancer datasets for organ-specific search. This indicates that LLMs' representations, although not perfect, can properly encode the text reports. However, one might question if this is due to the presence of medical keywords providing 'hints' to the model. To evaluate this, we removed the technical keywords such as cancer names and subtypes from the original text reports and then re-generated the representations with LLMs. The results can be found in Table 1b. As shown, the performance remains within the range of the organ-specific search. With this, we can empirically confirm that LLMs' representation is not only a function of the medical keywords but rather an encoding of the medical context. The same conclusion is held after investigating the results from the organ-independent search in Table 1c and 1d. We can potentially attribute this ability to the training scheme of language models as they are trained for next-token prediction for language modeling (Gloeckle et al., 2024; Achiam et al., 2023), meaning that they are not solely dependent on keywords or specific tokens in the text.

Furthermore, comparing 1a and Table 1c, we observe that LLMs maintain their performance when transitioning from organ-specific to organ-independent searches. This suggests that they are capable of encoding beyond specific information and semantically understanding the differences between reports from various organs and body parts. Among the different LLMs, it is particularly noteworthy that Bio-Llama3-8b almost always ranks among the top two best-performing models across various metrics, despite being significantly smaller in size compared to Command-R and Jamba. This highlights that smaller, bio-specific LLMs trained on medical data can encode pathology reports as effectively as larger, general-purpose LLMs.

### 4.3.2 SURVIVAL PREDICTION

For survival prediction, we used an ensemble Random Survival Forest (RSF) based on the implementation by (Ishwaran et al., 2008). This decision was due to the relative ease of implementation

Table 2: The C-Index of RSF for Survival Prediction on Eight Cancer Datasets using LLMs

| LLM | BRCA | GBM | KIRC | KIRP | LGG | LUAD | LUSC | UCEC |
|---|---|---|---|---|---|---|---|---|
| **Command-R** | $0.622_{\pm0.02}$ | $0.537_{\pm0.03}$ | $0.722_{\pm0.04}$ | $0.743_{\pm0.10}$ | $0.643_{\pm0.05}$ | $0.611_{\pm0.04}$ | $0.547_{\pm0.03}$ | $0.602_{\pm0.08}$ |
| **Gemma-7b** | $0.603_{\pm0.04}$ | $0.512_{\pm0.03}$ | $0.707_{\pm0.02}$ | $0.634_{\pm0.11}$ | $0.611_{\pm0.08}$ | $0.570_{\pm0.03}$ | $0.543_{\pm0.04}$ | $0.600_{\pm0.04}$ |
| **Jamba** | $0.625_{\pm0.05}$ | $0.501_{\pm0.02}$ | $0.689_{\pm0.06}$ | $0.745_{\pm0.09}$ | $0.639_{\pm0.06}$ | $0.595_{\pm0.06}$ | $0.545_{\pm0.01}$ | $0.617_{\pm0.07}$ |
| **Llama3-8b** | $0.629_{\pm0.05}$ | $0.521_{\pm0.04}$ | $0.713_{\pm0.03}$ | $0.759_{\pm0.07}$ | $0.607_{\pm0.07}$ | $0.585_{\pm0.07}$ | $0.520_{\pm0.06}$ | $0.580_{\pm0.08}$ |
| **Bio-Llama3-8b** | $0.627_{\pm0.07}$ | $0.537_{\pm0.03}$ | $0.709_{\pm0.05}$ | $0.726_{\pm0.08}$ | $0.587_{\pm0.07}$ | $0.583_{\pm0.03}$ | $0.548_{\pm0.04}$ | $0.621_{\pm0.04}$ |
| **Average** | $0.621_{\pm0.01}$ | $0.522_{\pm0.02}$ | $0.708_{\pm0.01}$ | $0.721_{\pm0.05}$ | $0.617_{\pm0.02}$ | $0.589_{\pm0.02}$ | $0.541_{\pm0.01}$ | $0.604_{\pm0.02}$ |

and training, as well as comparative robustness for higher input dimensionality. The RSF we used was from the scikit-survival python package (Pölsterl, 2020) and initialized with 1000 estimators. Afterward, it was trained using 5-fold cross-validation and 10 random seeds per fold. The evaluation metric we used was the right-censored concordance index, also known as the C–index.

We used the pathology report encodings generated by LLMs to train Random Survival Forests (RSFs) and report the C-index in Table 2. This clinically relevant task helps assess whether the LLM encodings capture features pertinent to patient outcomes. From a clinical perspective, cancer stage can help identify low- and high-risk patients for certain cancers. However, pathology reports contain far more detailed information, and this experiment demonstrates that LLMs are able to encode clinically relevant information in cancers such as BRCA, KIRC, KIRP, and UCEC, linking the encoded features to patient outcomes. To the best of our knowledge, this is the first time such a phenomenon has been observed. We have also compared LLMs with LVMs in this task in section H in Appendix.

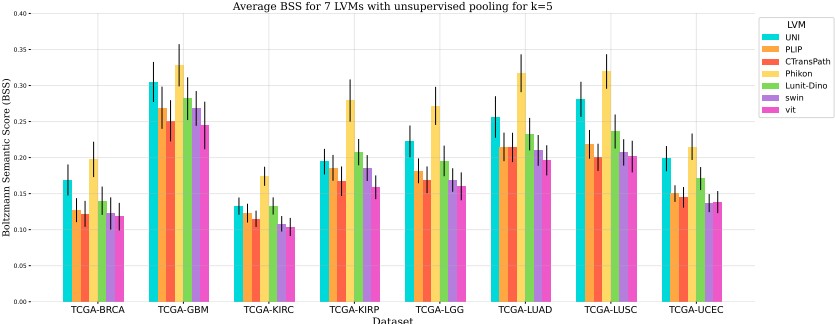

Figure 5: BSS comparison of seven LVMs using the average BSS across five LLMs on eight cancer datasets, with unsupervised pooling method, applied to all the cases in the dataset.

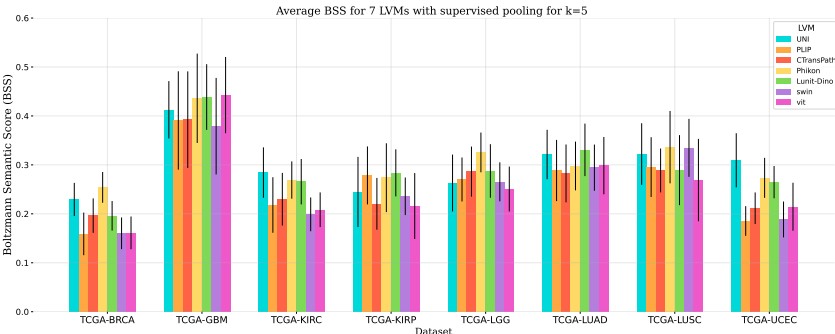

Figure 6: Average BSS comparison of seven LVMs using the average BSS across five LLMs on eight cancer datasets, with AbMIL as the supervised pooling method, applied to the cases in the held-out test set.

### 4.4 BOLTZMANN SEMANTIC SCORE

We have established a baseline for the current performance of LLMs in two tasks: information retrieval and survival prediction. Now, we experimentally evaluate and compare LVMs using the BSS, which leverages LLMs. In this study, we employ two widely used settings in digital pathology: unsupervised pooling and supervised pooling. For unsupervised pooling, we use mean pooling to represent each WSI based on its patches, while for supervised pooling, we trained a Multiple Instance Learning (MIL) model, specifically AbMIL (Ilse et al., 2018), following a 3-fold cross-validation scheme to represent the WSIs. We report the average BSS for $k = 5$ across five LLMs on eight TCGA datasets for both settings in Figure 5 and Figure 6. When comparing the two settings, we observe that supervised pooling improves the semantic quality of the features to some extent, but

it remains below $0.4$ for the majority of cancer datasets. More details regarding each setting are provided in section J.2 in Appendix.

Furthermore, LVMs achieve a fairly low BSS which indicates their encoding space may not be reflective of the information within a pathology report. The LVMs used were generally trained in a self-supervised manner. This means non-medically relevant features may be extracted, and contrastively clinically relevant features may be missed. Among pathology-pretrained models, UNI has not seen TCGA data, yet it outperforms the majority of the TCGA-pretrained LVMs. As UNI has been trained on 100,000 WSIs, compared to the TCGA-pretrained model with 12,000 WSIs, our observation suggests that models trained on more data may perform better semantically. As expected, ViT, which was trained on the non-medical ImageNet dataset, performed the worst overall. More details are provided in section D.3 and J in Appendix.

### 4.4.1 CORRELATION WITH DOWNSTREAM TASKS

Table 3: Two-sided Pearson Correlation Test between BSS and downstream task metrics

(a) Correlation between BSS and top-1 Accuracy in Information Retrieval

| LLM | GBM | | KIRC | | KIRP | | LGG | | LUAD | | LUSC | |
|---|---|---|---|---|---|---|---|---|---|---|---|---|
| | r | p-value | r | p-value | r | p-value | r | p-value | r | p-value | r | p-value |
| Command-R | 0.335 | $4.6e^{-03}$ | 0.605 | $2.9e^{-08}$ | 0.267 | $2.5e^{-02}$ | 0.274 | $2.2e^{-02}$ | 0.830 | $6.4e^{-19}$ | 0.557 | $5.4e^{-07}$ |
| Gemma-7b | 0.386 | $9.6e^{-04}$ | 0.483 | $2.3e^{-05}$ | 0.475 | $3.3e^{-05}$ | 0.309 | $9.2e^{-03}$ | 0.879 | $1.4e^{-23}$ | 0.459 | $6.4e^{-05}$ |
| Jamba | 0.302 | $1.1e^{-02}$ | 0.583 | $1.2e^{-07}$ | 0.247 | $4.0e^{-02}$ | 0.259 | $3.0e^{-02}$ | 0.838 | $1.5e^{-19}$ | 0.517 | $4.6e^{-06}$ |
| Llama3-8b | 0.293 | $1.4e^{-02}$ | 0.550 | $8.2e^{-07}$ | 0.326 | $5.9e^{-03}$ | 0.227 | $5.9e^{-02}$ | 0.807 | $3.3e^{-17}$ | 0.540 | $1.4e^{-06}$ |
| Bio-Llama3-8b | 0.355 | $2.6e^{-03}$ | 0.572 | $2.4e^{-07}$ | 0.323 | $6.5e^{-03}$ | 0.231 | $5.5e^{-02}$ | 0.834 | $2.9e^{-19}$ | 0.575 | $1.9e^{-07}$ |

(b) Correlation between BSS and C−index in Survival Prediction

| LLM | BRCA | | GBM | | KIRC | | KIRP | | LGG | | LUAD | | LUSC | | UCEC | |
|---|---|---|---|---|---|---|---|---|---|---|---|---|---|---|---|---|
| | r | p-value | r | p-value | r | p-value | r | p-value | r | p-value | r | p-value | r | p-value | r | p-value |
| Command-R | 0.344 | $1.9e^{-11}$ | −0.093 | $9.6e^{-01}$ | 0.273 | $1.1e^{-07}$ | 0.150 | $2.4e^{-03}$ | −0.027 | $6.9e^{-01}$ | 0.233 | $5.4e^{-06}$ | 0.099 | $3.2e^{-02}$ | 0.367 | $6.6e^{-13}$ |
| Gemma-7b | 0.307 | $2.3e^{-09}$ | −0.030 | $7.1e^{-01}$ | 0.245 | $1.8e^{-06}$ | 0.202 | $7.1e^{-05}$ | −0.017 | $6.2e^{-01}$ | 0.223 | $1.3e^{-05}$ | 0.081 | $6.6e^{-02}$ | 0.352 | $6.4e^{-12}$ |
| Jamba | 0.344 | $2.0e^{-11}$ | −0.080 | $9.3e^{-01}$ | 0.257 | $5.7e^{-07}$ | 0.144 | $3.4e^{-03}$ | −0.031 | $7.2e^{-01}$ | 0.209 | $3.9e^{-05}$ | 0.089 | $4.8e^{-02}$ | 0.352 | $5.7e^{-12}$ |
| Llama3-8b | 0.350 | $7.8e^{-12}$ | −0.085 | $9.4e^{-01}$ | 0.272 | $1.1e^{-07}$ | 0.151 | $2.4e^{-03}$ | −0.036 | $7.5e^{-01}$ | 0.218 | $1.9e^{-05}$ | 0.096 | $3.6e^{-02}$ | 0.370 | $4.4e^{-13}$ |
| Bio-Llama3-8b | 0.347 | $1.2e^{-11}$ | −0.087 | $9.5e^{-01}$ | 0.287 | $2.3e^{-08}$ | 0.131 | $7.1e^{-03}$ | −0.034 | $7.4e^{-01}$ | 0.224 | $1.1e^{-05}$ | 0.112 | $1.8e^{-02}$ | 0.373 | $2.9e^{-13}$ |

Beyond the average BSS and the insights it provides into understanding semantic capability, we also explored whether the score correlates with other metrics in downstream vision tasks, such as information retrieval and survival prediction.

Table 3a and 3b present the results of our hypothesis testing for the correlation between top-1 accuracy in information retrieval and BSS, as well as between the C-index for survival prediction and BSS. As shown, the correlation in information retrieval is higher than that in survival prediction. We attribute this to the fact that, in information retrieval, the representations remain unaltered for the task, whereas in survival prediction, the representations undergo RSF-supervised training, which adapts them to the task (in this case, patient risk), emphasizing certain features. Our findings show that, in certain cancer types, there are significant correlations between BSS and performance in both survival prediction and information retrieval. To the best of our knowledge, this is the first time it has been observed that structural similarity in one modality is correlated with the performance of a downstream task in another modality. For further explanations, please refer to section K in the Appendix.

### 4.4.2 CONSENSUS

One may argue that LLMs may not rank different LVMs similarly. However, to inspect this and measure agreement among various LLMs in ranking different LVMs based on BSS, Cohen's Kappa Score is utilized for pairwise comparison of rankings. According to this metric, a score ranging from $0.21$ to $0.40$ is considered fair, $0.41$ to $0.60$ is moderate, $0.61$ to $0.80$ is substantial, and $0.81$ to $1.00$ represents almost perfect agreement. In this test, LVM rankings produced by any LLM are compared across the 10 different configurations employed for the information retrieval task. As depicted in Figure 7, on average, a significant agreement is observed between Command-R and LLama3-8b; Command-R and Bio-Llama3-8b, Llama3-8b, and Jamba; and Llams3-8b and Bio-Llama3-8b, despite their complete independence and training on distinct data sources.

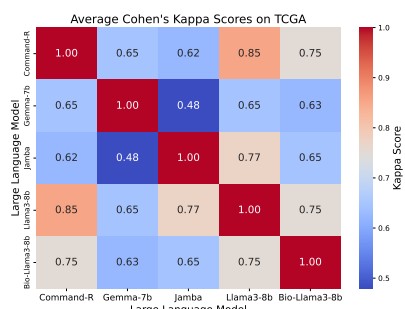

Figure 7: The average pairwise Cohen's Kappa Score between two LLMs on different cancer types in TCGA.

It is worth mentioning that moderate agreement, when present, depends on the dataset. For instance, on datasets such as GBM, we observe moderate agreement between LLMs, while on datasets such as KIRC and BRCA, there is significant agreement among the majority of LLMs. We attribute this inter-LLM variability on some datasets to the training data sources of these models. The pairwise Kappa matrix plots for each dataset are available in section N in the Appendix.

As a result, we observe a substantial overall agreement between LLMs, which we attribute to their use of a large dataset of reports in calculating the BSS, leading to greater consistency. Moreover, unlike qualitative approaches where agreement is difficult to measure, we can quantify it by comparing different LLMs and assessing the reliability of various sources. With the introduction of domain-specific LLMs, inter-LLM agreement has the potential to increase further, and BSS can provide a deeper and more specialized semantic measure.

## 5 DISCUSSION

The goal of this research was to introduce a systematic approach to measuring the semantic capability of LVMs. To achieve this, we first demonstrated the capability of LLMs in two key tasks in digital pathology: information retrieval and survival prediction. We then proposed the Boltzmann Semantic Score (BSS), a metric that measures structural similarity between two modalities, where we leverage LLMs to evaluate the semantic capability of LVMs. This metric provides quantifiable and collectively derived evidence of an LVM's performance, offering a more objective alternative to traditional small-scale qualitative assessments.

For the information retrieval task, we found that all LLMs achieved notable Top-1 performance, with Command-R attaining an average accuracy of 0.871 and an F1 score of 0.919 across all 32 pathology cancer datasets, encompassing $9,523$ patients. The results for the survival prediction task were more varied, with performance depending largely on the datasets and the clinical manifestations of the diseases. However, the C-index remained relatively consistent across the LLMs, with Command-R again showing the best average C-index of 0.628. Overall, we demonstrated, for the first time in the field, that LLM-derived features can be linked to patient outcomes, an area ripe for further exploration in future research.

Our next step was to assess the feasibility of the proposed Boltzmann Semantic Score for evaluating LVM performance. We observed that supervised pooling outperforms unsupervised pooling in WSI representation, largely due to the training step involved in supervised approaches. However, the semantic score for LVMs across different cancer datasets remains relatively low, suggesting there is considerable room for improvement in this area. We found that the BSS mostly correlated with the downstream task's performance metric, more so with retrieval than survival prediction. We attribute this difference to the methodological difference in the two tasks. In information retrieval, we use the exact same visual representations used for calculating BSS that were used for the task. However, in survival prediction, the visual representations undergo a regression step in the training phase that manipulates the representations, so more variations between BSS and C–index were expected.

We also conducted an experiment to investigate the consensus in LLMs when reporting their BSS for each LVM. We show that, first, LLMs are substantially in agreement on ranking the LVMs. Second, unlike qualitative approaches where agreement is not easily measurable; we can quantify it here by comparing different LLMs. With this, we expect with the rise of domain-specific LLMs, inter-LLM agreement even increases more.

## 6 CONCLUSION

Inspired by state space modeling, we have introduced the Boltzmann Semantic Score (BSS) as a novel method for leveraging the encoding space of LLMs to evaluate the encoding space of LVMs. We first evaluated five state-of-art LLMs in clinically inspired digital pathology tasks and showed (for the first time) their capability in predicting patient outcomes. We then measured the BSS associated with seven LVMs using each of the five LLMs and a large collective database of pathology reports. Overall, LVMs showed poor BSS, highlighting their low semantic capability. We also found that the BSS for the seven evaluated LVMs is highly correlated with performance in two clinical tasks: information retrieval and survival prediction.

## 7 ACKNOWLEDGMENT

We would like to thank Nikolay Alabi for validating the qualitative findings presented in this work. We also acknowledge the use of computational resources provided by Canada's Michael Smith Genome Sciences Centre (GSC) at BC Cancer, which supported our experiments and analyses. Additionally, we appreciate the insightful discussions and feedback from our colleagues and mentors, which have contributed to refining our methodology and interpretations.

## 8 REPRODUCIBILITY STATEMENT

We believe that the theory proposed here for the Boltzmann Semantic Score is not limited to the medical field and can be applied to other domains with similar contexts. Therefore, the code developed and the processed data will be publicly released in our GitHub repository. We have also listed all the packages that have been used in section E in the Appendix. Besides the theoretical discussion in the paper, we have provided the algorithms in Algorithm 1, 2, and 3. For all the experiments, the results or the dataset description has been written in detail in the corresponding section in the Appendix.

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

**Appendix**

Below is an outline of the appendix for easy reference:

Table 4: 32 TCGA cancer datasets with their full names and number of cases used in our study

| Abbreviation | Full Name | Number of Patients |
|---|---|---|
| ACC | Adrenocortical Carcinoma | 90 |
| BLCA | Bladder Urothelial Carcinoma | 379 |
| BRCA | Breast Invasive Carcinoma | 1034 |
| CESC | Cervical Squamous Cell Carcinoma and Endocervical Adenocarcinoma | 289 |
| CHOL | Cholangiocarcinoma | 43 |
| COADREAD | Colon Adenocarcinoma/Rectum Adenocarcinoma | 580 |
| DLBC | Lymphoid Neoplasm Diffuse Large B-cell Lymphoma | 47 |
| ESCA | Esophageal Carcinoma | 105 |
| GBM | Glioblastoma Multiforme | 293 |
| HNSC | Head and Neck Squamous Cell Carcinoma | 520 |
| KICH | Kidney Chromophobe | 112 |
| KIRC | Kidney Renal Clear Cell Carcinoma | 525 |
| KIRP | Kidney Renal Papillary Cell Carcinoma | 280 |
| LGG | Brain Lower Grade Glioma | 575 |
| LIHC | Liver Hepatocellular Carcinoma | 341 |
| LUAD | Lung Adenocarcinoma | 353 |
| LUSC | Lung Squamous Cell Carcinoma | 318 |
| MESO | Mesothelioma | 79 |
| OV | Ovarian Serous Cystadenocarcinoma | 371 |
| PAAD | Pancreatic Adenocarcinoma | 176 |
| PCPG | Pheochromocytoma and Paraganglioma | 174 |
| PRAD | Prostate Adenocarcinoma | 446 |
| SARC | Sarcoma | 249 |
| SKCM | Skin Cutaneous Melanoma | 102 |
| STAD | Stomach Adenocarcinoma | 319 |
| STES | Stomach and Esophageal Carcinoma | 83 |
| TGCT | Testicular Germ Cell Tumors | 87 |
| THCA | Thyroid Carcinoma | 487 |
| THYM | Thymoma | 114 |
| UCEC | Uterine Corpus Endometrial Carcinoma | 546 |
| UCS | Uterine Carcinosarcoma | 56 |
| UVM | Uveal Melanoma | 65 |
| Sum | | 9,238 |

## A    STATEMENT ON LIMITATIONS

Bias can be carried over both from text and image in the dataset, this can be variations within text reports originating from different hospitals, doctors, and cohorts or on WSIs from using different staining protocols, hospitals, and cohorts. As BSS relies on LLMs for representing text reports, based on their capability to extract information from textual data, we expect some robustness to dataset-originating biases. However, this needs to be further evaluated and also with newer LLMs, it can be improved if it is proved to be biased. Therefore, other bias metrics should complement BSS for a more comprehensive assessment of dataset-originating biases.

Our experiments have been limited to two specific tasks: information retrieval and survival prediction. Therefore, further studies are necessary to evaluate the generalizability of our approach to other medically relevant tasks.

## B    STATEMENT ON GENERALIZATION

BSS generalization in the medical domain depends on the nature of the task. For tasks directly linked to textual information, such as disease classification or subtype detection, we expect strong generalization as TCGA is made up of data from multiple clinical sites, which may have different reporting standards and procedures for what information is included in the pathology reports. For complex tasks like survival or treatment prediction, where insights are not explicitly available in reports, generalization may vary depending on the data as well as the patient cohorts, necessitating further experimentation. Additionally, tasks like segmentation, with limited text-image linkage, are less likely to generalize and depend on specific data context, which can vary by cancer type. In addition, while our study focuses on text and image modalities, the BSS framework can extend to other modality pairs (e.g., genomics and imaging), using genomics as a reference for biological semantics. We anticipate that BSS may generalize to other fields beyond medicine, though further validation is needed.

## C  Statement on Future Directions

We believe it is essential to develop approaches that incorporate semantic representation at a higher level. While current vision-language models focus primarily on patch-level representations, there is a need for slide-level semantic representation approaches to advance the field. In terms of optimization, BSS offers a promising avenue as it is differentiable and can be leveraged as a loss function to transfer semantic knowledge from text to vision models. By integrating BSS into training pipelines, vision models can better align with the semantic structures inherent in textual data. Additionally, this concept can be combined with contrastive learning, where positive and negative samples are defined based on BSS. Such a combination could refine model training by encouraging the alignment of semantically similar pairs while distinguishing dissimilar ones. This approach could help bridge the gap between text and vision modalities, leading to more robust and semantically-aware models for tasks requiring holistic understanding at the slide level.

## D  Dataset

We used preprocessed reports from (Kefeli & Tatonetti, 2024). Briefly, Kefeli et al. utilized OCR to convert PDF reports to text. Additionally, identifiable information (e.g., patient IDs, hospital names, pathologists) was removed by them, and the data was validated through a proof-of-principle cancer-type classification task across 32 tissues, achieving an average AU-ROC of 0.992. Further details can be found in (Kefeli & Tatonetti, 2024). Here, we list the cancer types we used for different experiments in our work. The number of patients is the same as mentioned in (Kefeli & Tatonetti, 2024).

### D.1  Information Retrieval

For the information retrieval task, we used two settings. For the organ-independent search, we used all the cancer types present in Table 4. In this setting, we query one of the cancer types among all the 32 cancer types.

For the organ-specific search, we distributed the cancer types as demonstrated in Table 5 into 9 different body organs. Then, when searching, we only queried among the cancer type associated with the same body organ.

Table 5: Organs and their associated cancers for organ-specific search in TCGA

| Organ | Associated Cancers |
|---|---|
| Brain | GBM, LGG |
| Endocrine | ACC, PCPG, THCA |
| Gastro | COADREAD, ESCA, STAD |
| Gynaeco | CESC, OV, UCS, UCEC |
| Liver-Panc | CHOL, LIHC, PAAD |
| Melanocytic | SKCM, UVM |
| Prostate-Testis | PRAD, TGCT |
| Pulmonary | LUAD, LUSC, MESO, THYM |
| Urinary | BLCA, KICH, KIRC, KIRP |

### D.2  Survival Prediction

For the survival prediction task, we used BRCA, GBM, KIRC, KIRP, LGG, LUAD, LUSC, and UCEC datasets from TCGA.

### D.3  Boltzmann Semantic Score

For calculating the Boltzmann Semantic Score we used BRCA, GBM, KIRC, KIRP, LGG, LUAD, LUSC, and UCEC from TCGA. However, for the correlation test with the information retrieval task, we divided these eight cancers into groups associated with their organs for conducting the retrieval. This is because, for a retrieval task, we at least need two classes to calculate accuracy, F1 score, and mean average precision score. Therefore, the groups were as follows: GBM and LGG; KIRC and

KIRP; and LUSC and LUAD. As BRCA and UCEC cannot be associated with another cancer in the 8 selected TCGA datasets, we put both aside for this task. For the correlation test with the survival prediction task, we used the eight cancer sets individually and reported the results.

Along with the paper, we have uploaded the cancer by cancer Boltzmann Semantic Score for each dataset for all the $k$s in $\{1, 3, 5, 20, 50\}$ in the supplementary materials. Please note that the error bars in the plots are **the standard deviation of each LVM over all the patients present in that cancer type**. Thus, error bars are huge for small $k$s which shows the intense variability of LVMs. However, as $k$ increases, the average Boltzmann Semantic Score increases and the variability reduces, which is due to the larger state spaces making it easier for the LVMs to find matches. We believe $k = 5$ is a good choice that provides a larger state space without losing sensitivity.

Please note that the supplementary materials is organized in a way such that the directory names are the cancer types, and the name of each '.png' file shows the name of the file with full details. For example, *'/BRCA/top-1_Coral-Gemma-7b-Jamba-Llama3-8b-Bio-Llama3-8b.png'* shows that it belongs to BRCA cancer, and *top-1* shows that it is the average BSS calculated based on $k = 1$. Please see *'Boltzmann Semantic Score.zip'* to download all related files.

## E    Software and Packages

We utilized an array of different Python packages to conduct data processing and visualization tasks. The data manipulation and analysis were facilitated using Pandas (version 1.5.3) and NumPy (version 1.23.5), while statistical computations were performed using SciPy (version 1.10.1). For machine learning and deep learning models, we employed Scikit-Learn (version 1.2.1), OpenCV (cv2) (version 4.7.0), TorchVision (version 0.14.1), and PyTorch (version 2.2.1+cu121), respectively. Visualization of data and results was achieved through Matplotlib (version 3.7.0) and Seaborn (version 0.12.2). Additionally, we leveraged the Transformers library (version 4.31.0) for natural language processing tasks and the Timm library (version 0.9.16) for working with some of the vision models.

## F    LLM Representations

We use the Huggingface implementations of the LLMs and their default tokenizers to extract the representations. For each LLM, the representations from the last hidden state were extracted and then mean-pooled over all the tokens to obtain the report-level representation tensor for each report.

Table 6: Representation dimensions for different LLMs.

| LLM | Jamba | Llama3-8b | Bio-Llama3-8b | Gemma-7b | Command-R |
|---|---|---|---|---|---|
| $d_1$ | 1024 | 4096 | 4096 | 3072 | 8192 |

## G    LVM Representations

First, we tile the Whole Slide Images (WSIs) at 20x microscopic magnification into patches of size $224 \times 224$. These patches are then passed through the LVM to obtain patch-level representations. Subsequently, all the patch-level representations are pooled by both supervised and unsupervised pooling methods to obtain the WSI-level representation tensor. If a patient has more than one WSI, we average the WSI-level representations to derive the patient-level representation.

Table 7: Representation dimensions for different LVMs.

| LVM | ViT | SwinT | PLIP | CTransPath | Lunit-Dino | Phikon | UNI |
|---|---|---|---|---|---|---|---|
| $d_2$ | 768 | 1024 | 512 | 768 | 384 | 768 | 1024 |

## H    Survival Prediction Experiments

### H.1    A discussion on TCGA-GBM Survival Prediction

In Table 2, the TCGA-GBM dataset showed the lowest average performance in the survival prediction task on LLM features, with a C-index of 0.522. Biologically, this aligns with the aggressive and

heterogeneous nature of glioblastoma multiforme, a class of highly malignant brain tumors. The TCGA-GBM cohort has a median survival of only 12.23 months, with approximately half of the patients either deceased or lost to follow-up within the first year. Glioblastomas are usually detected in later stages and there is a lack of robust prognostic biomarkers in tissue images for patient risk stratification. Moreover, two of the most informative prognostic factors—mutations in IDH and MGMT genes —are genomic variables that may be inaccessible to LLMs through pathology reports, potentially explaining the gap in performance.

## H.2 LVM's Survival Prediction

We used the pathology reports encodings of LLMs to train RSFs and report the C–index in Table 2. This is a clinically relevant task to compare the LLMs and LVMs. As we are using zero-shot LLMs here, Vit and SwinT are probably the best comparisons for a zero-shot image encoder, as they were trained on natural, non-medical images. Overall, the RSF models trained on LLM representations outperformed those trained on LVM features as shown in Table 2 and 8. It is worth noting that CTransPath, Lunit-Dino, and Phikon were trained on TCGA data, and thus may be better able to extract representative imaging features. However, for some cancers like GBM and LGG, pathology-pretrained LVMs are performing better than LLMs. With that being said, LLMs' performance on average is better than LVMs even though they have not been trained on pathology data.

Table 8: The C-Index of RSF on Eight Cancer Types using LVMs

| LVM | BRCA | GBM | KIRC | KIRP | LGG | LUAD | LUSC | UCEC |
|---|---|---|---|---|---|---|---|---|
| ViT | $0.579_{\pm0.05}$ | $0.587_{\pm0.02}$ | $0.655_{\pm0.05}$ | $0.624_{\pm0.12}$ | $0.698_{\pm0.06}$ | $0.567_{\pm0.02}$ | $0.572_{\pm0.05}$ | $0.628_{\pm0.11}$ |
| SwinT | $0.564_{\pm0.04}$ | $0.577_{\pm0.01}$ | $0.669_{\pm0.05}$ | $0.624_{\pm0.07}$ | $0.687_{\pm0.02}$ | $0.543_{\pm0.03}$ | $0.559_{\pm0.04}$ | $0.604_{\pm0.11}$ |
| CTransPath | $0.622_{\pm0.07}$ | $0.565_{\pm0.05}$ | $0.703_{\pm0.05}$ | $0.685_{\pm0.03}$ | $0.706_{\pm0.09}$ | $0.582_{\pm0.04}$ | $0.568_{\pm0.02}$ | $0.678_{\pm0.03}$ |
| Lunit-Dino | $0.589_{\pm0.04}$ | $0.563_{\pm0.03}$ | $0.682_{\pm0.02}$ | $0.742_{\pm0.07}$ | $0.708_{\pm0.07}$ | $0.587_{\pm0.04}$ | $0.574_{\pm0.04}$ | $0.661_{\pm0.06}$ |
| PLIP | $0.575_{\pm0.04}$ | $0.541_{\pm0.02}$ | $0.665_{\pm0.03}$ | $0.706_{\pm0.15}$ | $0.694_{\pm0.04}$ | $0.569_{\pm0.06}$ | $0.545_{\pm0.04}$ | $0.643_{\pm0.09}$ |
| Phikon | $0.626_{\pm0.03}$ | $0.582_{\pm0.04}$ | $0.711_{\pm0.05}$ | $0.736_{\pm0.04}$ | $0.684_{\pm0.04}$ | $0.593_{\pm0.03}$ | $0.545_{\pm0.03}$ | $0.696_{\pm0.06}$ |
| UNI | $0.634_{\pm0.05}$ | $0.590_{\pm0.03}$ | $0.715_{\pm0.03}$ | $0.771_{\pm0.06}$ | $0.729_{\pm0.04}$ | $0.564_{\pm0.04}$ | $0.582_{\pm0.05}$ | $0.697_{\pm0.05}$ |
| Average | $0.598_{\pm0.04}$ | $0.572_{\pm0.03}$ | $0.686_{\pm0.04}$ | $0.698_{\pm0.08}$ | $0.701_{\pm0.05}$ | $0.572_{\pm0.04}$ | $0.563_{\pm0.04}$ | $0.658_{\pm0.07}$ |

## I Ablation Study on Patches

As an ablation study, we extracted a varying number of patches, ranging from 150 to 1500, from WSIs and mean-pooled them to obtain the slide-level representations. Afterward, we calculated the average BSS over each cancer dataset using different $k$'s in $\{1, 3, 5, 20, 50\}$. Figure 8 depicts the average BSS on the eight cancer datasets of TCGA and the five LLMs for each vision model for $k = 5$. As can be seen, the score is stable over different numbers of patches, which means that more patches do not guarantee a better semantic representation.

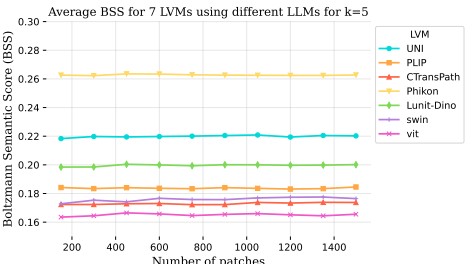

Figure 8: The average BSS with $k = 5$ over different LLMs and different cancer types in TCGA.

## J Boltzmann Semantic Score Algorithm

We discussed the mathematical way of obtaining BSS, here we provide Algorithm 1 that demonstrates the mechanism to calculate $\mathcal{B}_q$ for the query $q$. Please note that the BSS is the average of all $\mathcal{B}_q$'s over one cancer type or class.

## J.1 Computational Complexity

If we consider a database of size $n$, the order of complexity for KNN is $O(nd + n \log n)$, where $d$ is the feature size. In Algorithm 1, we rely on graphs with $k$ neighbors obtained from KNN. To calculate the second-order Boltzmann factor, the order of complexity is $O(k^2)$. Therefore, the total order of complexity, including the KNN algorithm in our calculations, is $O(nd + n \log n + k^2)$. However, $k$ is a constant that we chose to be $k = 5$. Hence, $k^2$ is negligible compared to $nd$ or $n \log n$ from the KNN algorithm. Consequently, the order of complexity remains that of the KNN

---

**Algorithm 1** Boltzmann Semantic Score Algorithm

---

**Require:** $reference\_graph$, $observer\_graph$, $d_1$
**Ensure:** score $B_q$
 1: Initialize $total\_cost$, $non\_matching\_nodes\_cost$ to 0
 2: Initialize node index position counter
 3: **for** each node in $text\_graph$ excluding the query node **do**
 4:     Calculate Boltzmann Factor from current node to reference node in $reference\_graph$
 5:     **if** current node is not estimated by $observer\_graph$ **then**
 6:         Estimate corresponding node in $observer\_graph$
 7:         Calculate the Second-order Boltzmann Factor from the current and estimated nodes
 8:         Update $lost\_nodes\_cost$ with computed Second-order Boltzmann Factor
 9:     **end if**
10:     Update $total\_cost$ with computed Boltzmann Factor or Second-order Boltzmann Factor
11: **end for**
12: Compute $B_q$ by combining $lost\_nodes\_cost$ and $extra\_nodes\_cost$
13: **return** $B_q$

---

algorithm, which is $O(nd + n \log n)$. We have explored this with $k$ values ranging from 1 to 80, without any computational difficulty on our low RAM CPUs.

### J.2 BOLTZMANN SEMANTIC SCORE EXPERIMENTS

For the unsupervised pooling, since no training was required, we tested the model on the entire dataset and reported the results. However, for the supervised pooling, we trained the AbMIL model on the features extracted by each LVM. To ensure a robust evaluation, we employed a 3-fold cross-validation scheme to train, validate, and test the model, leaving samples from one patient all on one set to prevent data leakage. We repeated the experiments with 10 different random seeds and selected the top-performing seed for the finalized model. Afterward, we calculated the BSS using the output of the AbMIL model as the WSI representation. For training, we used an eight-class classifier over the eight cancer datasets, with a learning rate of 0.001, weight decay of 0.01, and trained for 25 epochs. Patches were extracted from $20\times$ magnifications.

## K CORRELATION WITH DOWNSTREAM TASKS

We use the two-sided Pearson correlation test, where the null hypothesis ($H_0$) states that there is no significant correlation between the variables, and the alternative hypothesis ($H_1$) states that there is a significant correlation. Here, we have two tasks: information retrieval and survival prediction.

A separate Random Survival Forest was trained on each of the visual embedding sets. Each forest was built with 1000 estimators, and followed the algorithm (implemented using scikit-survival library) as described in Ishwaran et al (Ishwaran et al., 2008). A 5-fold cross-validation strategy was used such that for each fold a fifth of the data was held out as the test set. The remaining data was organized into 10 random 80/20 splits of training and validation data. Thus for each dataset, we had a total of $10 \times 5 = 50$ test C–index values. As such, we then had a total of $7 \times 50 = 350$ C–index samples for the 7 pathology datasets considered in this paper.

For each dataset, the Boltzmann Semantic Score (BSS) was calculated. We then used the two-sided Pearson correlation test to find the correlation between the BSS values and the C–index samples for each dataset. The p-value and correlation for each were reported in the body of this paper, with a p–value $< 0.05$ considered significant. Algorithm 2 describes the correlation test with survival prediction experiment accurately.

For each dataset, the Boltzmann Semantic Score (BSS) was calculated. We then used the Pearson correlation test to find the correlation between the BSS values and the top-k accuracy samples for each dataset. The p-value and correlation for each were reported in the body of this paper, with a p–value $< 0.05$ considered significant.

We adopt the above sampling approach to gather enough random samples for the Pearson correlation test so that the results of the test are reliable. Algorithm 3 describes the correlation test with information retrieval experiment accurately.

---

**Algorithm 2** Correlation Test with Survival Prediction

---

**Require:** Visual embedding sets $\{LVM_1, LVM_2, \ldots, LVM_7\}$ of a given dataset
 1: Initialize number of estimators $N_{\text{estimators}} \leftarrow 1000$
 2: Initialize number of folds $T \leftarrow 5$
 3: Initialize number of random seeds $N_s \leftarrow 10$
 4: **for** each visual embedding set $LVM_i$ **do**
 5:     **for** each random seed $s \in \{s_1, \ldots, s_{N_s}\}$ **do**
 6:         **for** each fold $t \in \{1, \ldots, T\}$ **do**
 7:             Split $LVM_i$ into test set $T_t$ (20% of data) and remaining data $R_k$ (80% of data)
 8:             Train Random Survival Forest
 9:             Evaluate model on $T_t$ and obtain test C–index value $C_{i,t,s}$
10:             Compute average Boltzmann Semantic Score (BSS) $B_{i,t,s}$
11:         **end for**
12:     **end for**
13: **end for**
14: Collect all the C–index samples in a list Collect all the BSS samples in a list
15: Calculate Pearson correlation between BSS list and C–index list
16: Perform two-sided Pearson correlation test and obtain p-value
17: **if** p-value $< 0.05$ **then**
18:     Report correlation and p-value as significant
19: **else**
20:     Report correlation and p-value as not significant
21: **end if**
22: Report correlations and p-values for the dataset

---

### K.1   Interpretation of Hypothesis Testing

The hypotheses defined in this study provide statistical proof and evidence that the Boltzmann Semantic Score (BSS) is predictive for downstream tasks. Ideally, if an LVM captures a high semantic score, it is likely to perform well on downstream tasks, though the relationship is not completely linear. Therefore, we assert that when BSS is significantly correlated with Accuracy and/or the C–index in information retrieval and/or survival prediction, it indicates two key points: first, the correlation is not a random event, and second, for the specific cancer data studied, LVMs with higher BSS are generally expected to perform better on these tasks, yet proportional to the correlation.

The variations in correlation observed in Table 3a for BSS with information retrieval top-1 accuracy across different datasets and LLMs can be attributed to several factors. Firstly, the dataset-specific differences, such as the type and quality of information present in pathology reports, play a significant role. In some datasets, like TCGA-LUAD, we observe that LVMs with higher BSS values tend to align better with LLM semantic spaces, resulting in stronger correlations with information retrieval performance. This indicates a stronger semantic alignment with the LLM's semantic space in this dataset.

However, in datasets like TCGA-LGG, the correlation is weaker, potentially due to LVMs retrieving cases that are less semantically aligned yet still achieving high top-1 accuracy. This can occur because accuracy, when measured solely by subtype, may overlook nuances in semantic similarity that BSS captures. As a result, some LVMs might succeed in retrieving correct subtypes but fail to retrieve semantically similar cases, leading to a drop in correlation.

Secondly, the latent spaces of different LLMs have subtle distinctions, influencing the correlation values observed within each cancer dataset. These variations in LLM latent spaces likely contribute to the differing degrees of alignment and correlation across datasets, which matches our expectations given the inherent complexity and context-specific nuances in each dataset.

Similarly, for the information retrieval task, we conducted the search and calculated the top-1, top-3, and top-5 accuracy with slide-level representation derived from varying patch numbers ranging from 150 to 1500 with a step of 100. This allowed us to gather 10 samples per each dataset. As such, we then had a total of $7 \times 10 = 70$ samples for each of the top-k accuracy metrics.

---

**Algorithm 3** Correlation Test with Information Retrieval

---

**Require:** Patch numbers $P \leftarrow \{150, 250, 350, \dots, 1500\}$ for a given dataset
 1: Initialize top-k metrics $K \leftarrow \{1, 3, 5\}$
 2: **for** each patch number $p \in P$ **do**
 3:     Compute slide-level representation for $D_i$ using $p$ patches
 4:     **for** each top-k metric $k \in K$ **do**
 5:         Conduct the search and Calculate top-$k$ accuracy $A_{i,p,k}$
 6:         Compute average Boltzmann Semantic Score (BSS) $B_{i,p,k}$
 7:     **end for**
 8: **end for**
 9: Collect all the top-$k$ accuracy samples in a list Collect all the BSS samples in a list
10: Calculate Pearson correlation between BSS list and top-$k$ accuracy list
11: Perform two-sided Pearson correlation test and obtain p-value
12: **if** p-value $< 0.05$ **then**
13:     Report correlation and p-value as significant
14: **else**
15:     Report correlation and p-value as not significant
16: **end if**
17: Report correlations and p-values for the dataset

---

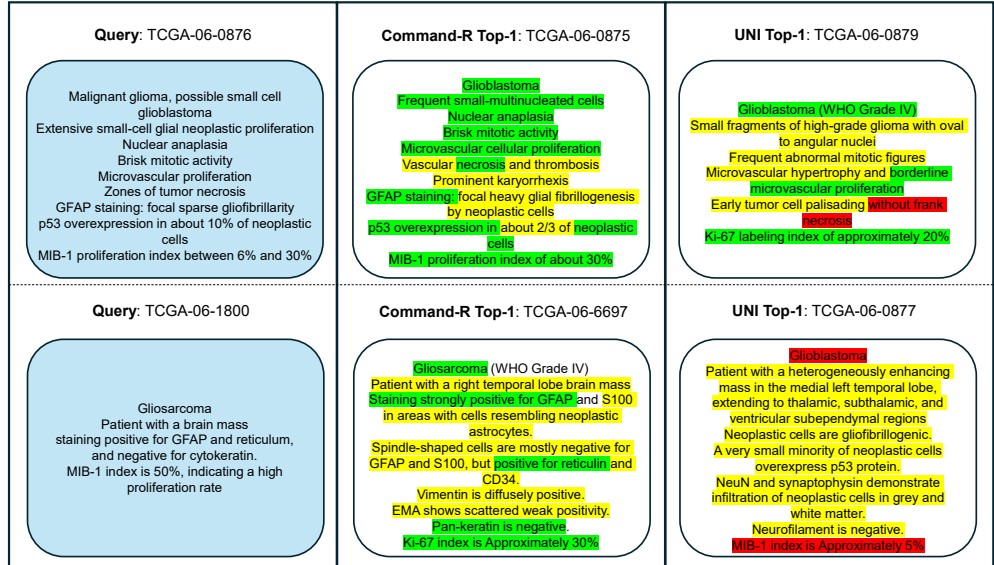

Figure 9: Two random queries from the TCGA-GBM dataset are shown. In this experiment, we reported the top-1 retrieved case for Command-R as a candidate from LLMs and for UNI as a candidate from LVMs. By querying the case, we summarized the key microscopic details in the pathology reports here. The texts highlighted in green are semantically similar to the details in the reports; yellow indicates extra information mentioned in the retrieved case that is not fundamentally different; and red indicates semantically different information, making the case dissimilar to that of the query. In both cases, Command-R retrieved a more semantically similar case than UNI, although the cases retrieved by UNI had some similarities.

## L QUALITATIVE EXAMPLES

Two random queries from the TCGA-GBM dataset are shown in Figure 9.

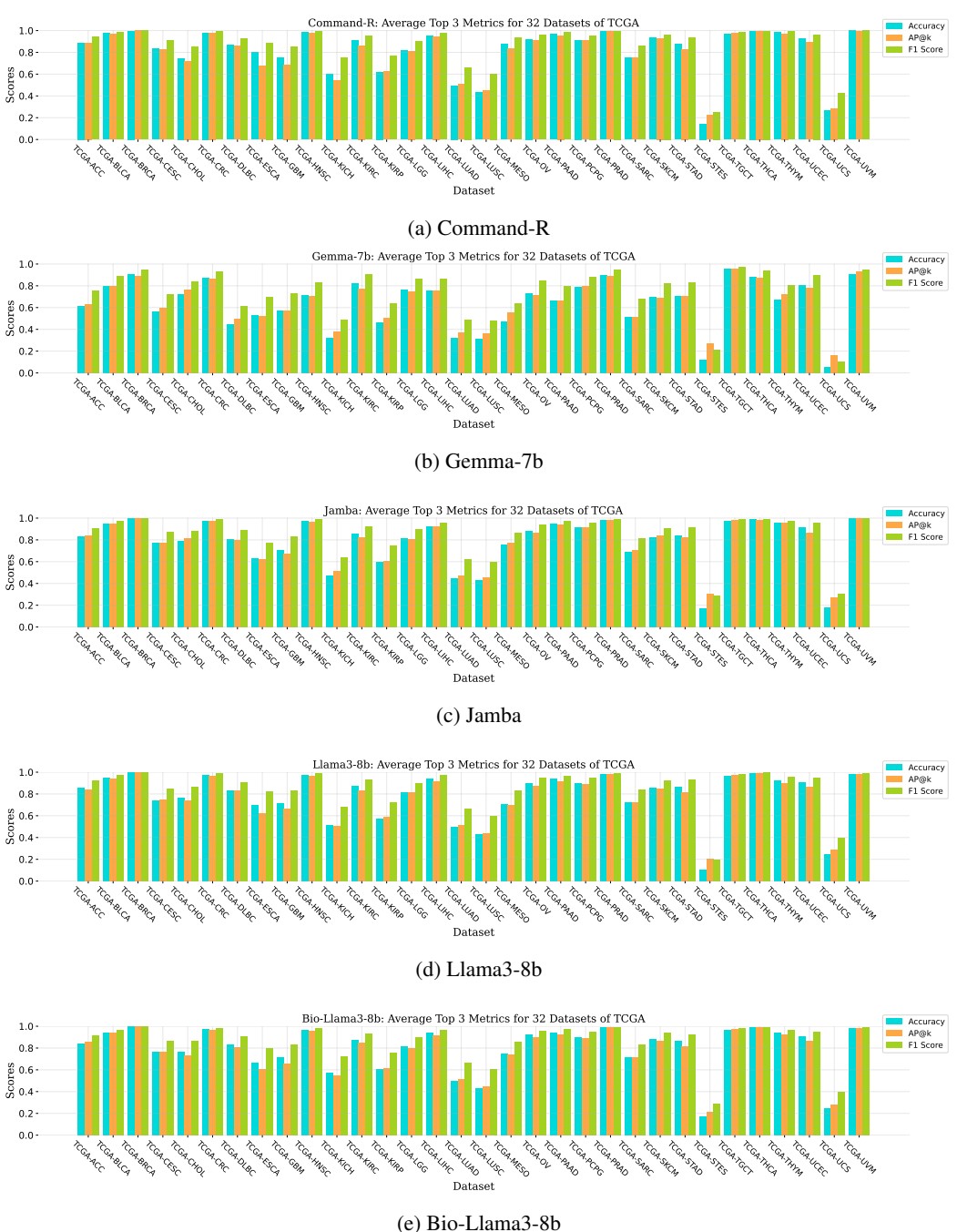

Figure 10: Organ-Independent search on the original text reports

## M   LLM REPRESENTATION PLOTS

The LLM performance on each cancer dataset through organ-specific and organ-independent for two scenarios of original and perturbed text reports for different LLMs are shown in Figure 10, 11, 12, and 13.

## N   CONSENSUS: CANCER BY CANCER

Cancer by cancer Cohen's Kappa Score is illustrated in Figure 14. As discussed in the body of paper, we know that the agreement between different LLMs varies cancer by cancer. Here, on LGG, KIRC,

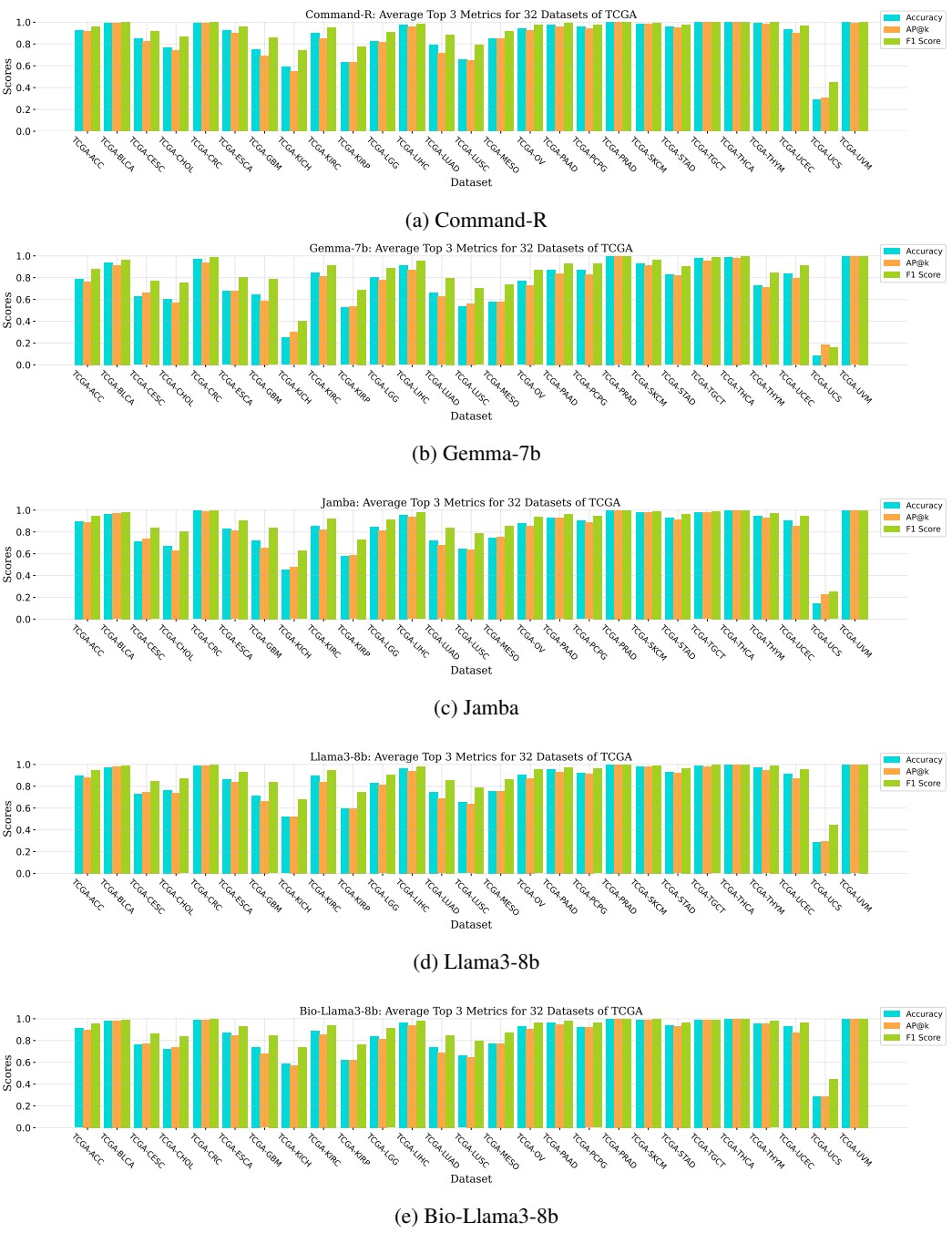

Figure 11: Organ-specific search on the original text reports

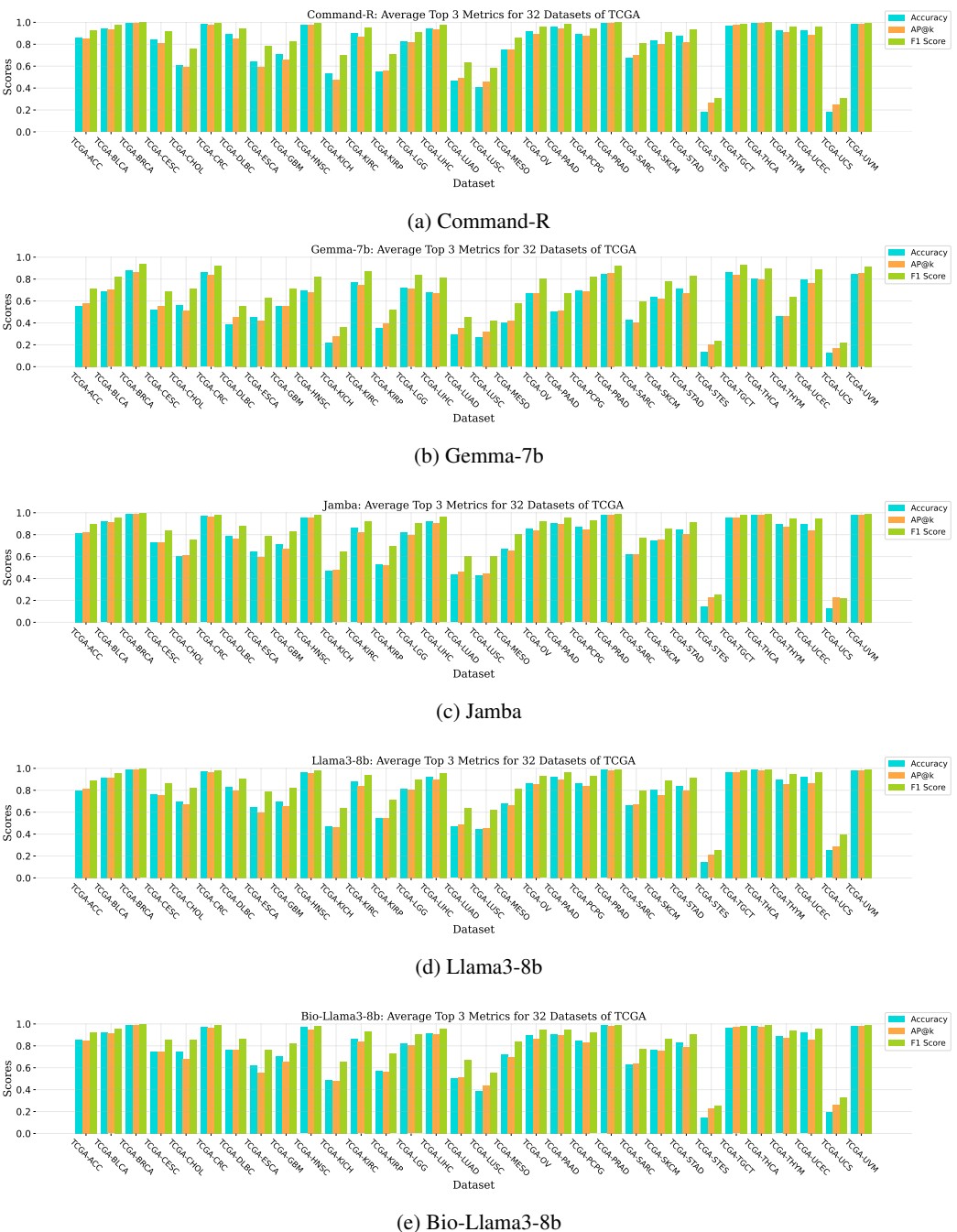

Figure 12: Organ-independent search on the perturbed text reports

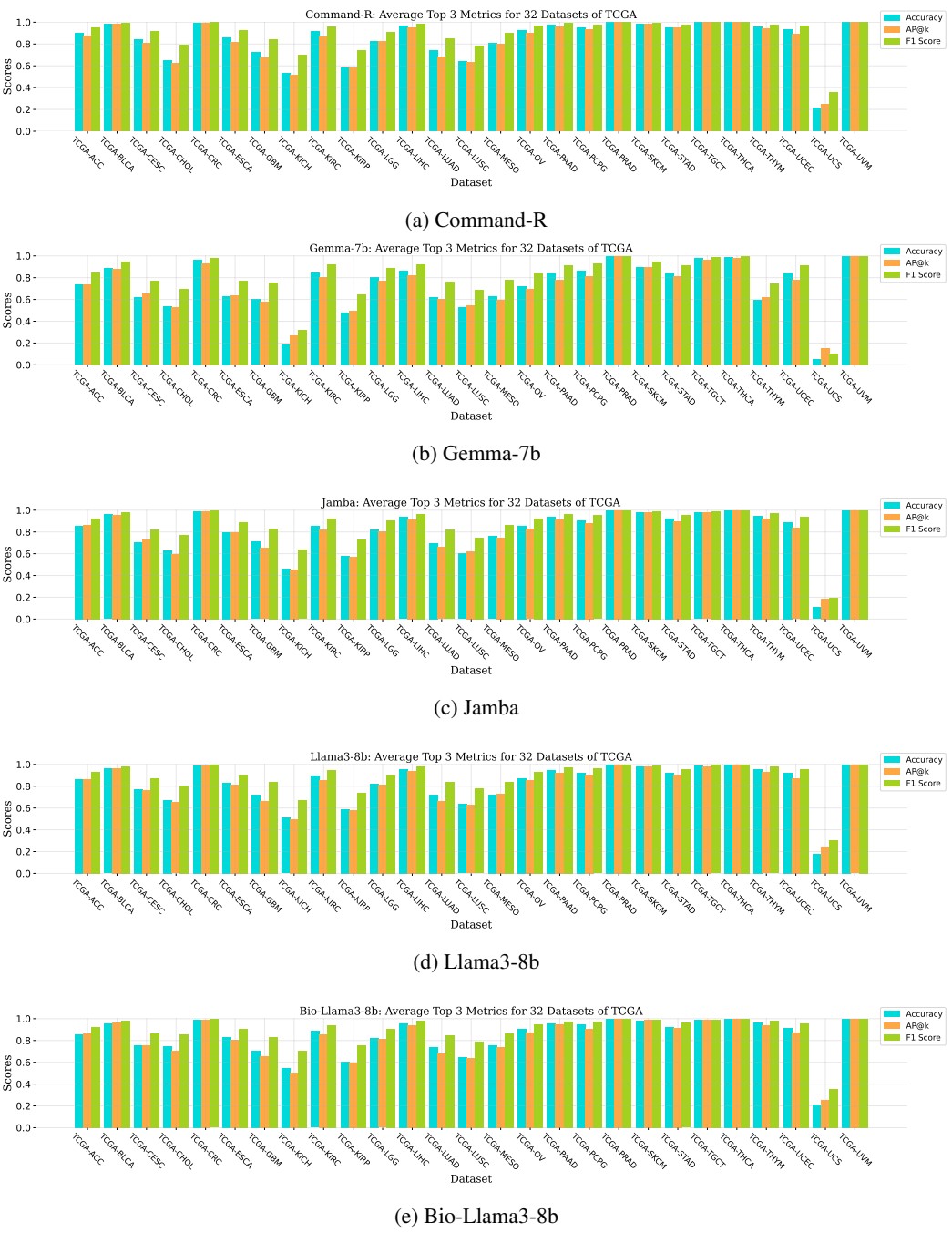

(a) Command-R

(b) Gemma-7b

(c) Jamba

(d) Llama3-8b

(e) Bio-Llama3-8b

Figure 13: Organ-specific search on the perturbed text reports

LUAD, LUSC, and BRCA there are significant agreement between LLMS. Yet, on UCEC, KIRP, and GBM, LLMs do not similar agreement in ranking different LVMs.

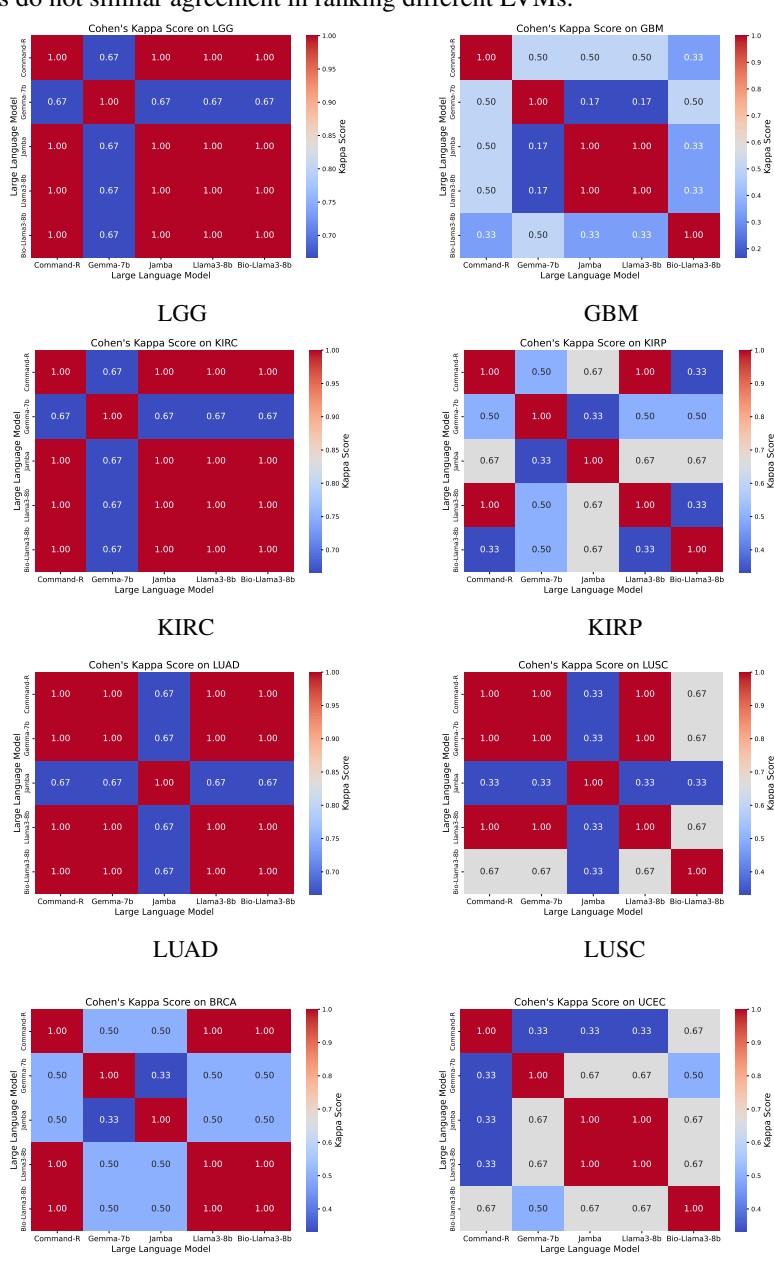

Figure 14: Cancer by Cancer Cohen's Kappa Score. As seen, on LGG, KIRC, LUAD, LUSC, and BRCA, there are significant agreements between LLMS. Yet, on UCEC, KIRP, and GBM, LLMs tend to disagree. According to this metric, a score ranging from $0.21$ to $0.40$ is considered fair, $0.41$ to $0.60$ is moderate, $0.61$ to $0.80$ is substantial, and $0.81$ to $1.00$ represents almost perfect agreement.

