# OpenReview forum: "Boltzmann Semantic Score: A Semantic Metric for Evaluating Large Vision Models Using Large Language Models"
_ICLR.cc/2025/Conference — ICLR 2025 Poster_

### Official Review · Reviewer_F61b · 2024-10-28

**Soundness:** 4
**Presentation:** 3
**Contribution:** 4
**Rating:** 8
**Confidence:** 3

**Summary:**

This paper proposes a semantic metric, BBS, to evaluate LVMs from a medically semantic perspective.

The paper also leverages LLMs and a large and collective database of medical reports across more than 30 cancer types that represent more than 9,500 patients and it also establishes a baseline of LLMs' performance in two large-scale digital pathology tasks.

**Strengths:**

* Originality: Noval evaluation metric which evaluates the encoding space of LVMs from medical images using the encoding space of Large Language Models (LLMs) from medical reports.

* Quality: Extensive experiments including experimentation on 32 datasets from The Cancer Genome Atlas collection using five state-of-the-art LLMs, comparison of seven LVMs with BSS, and two correlation analyses between BSS and performance in two downstream tasks.

* Clarity: Well-painted figures and clear formulas.

* Significance: Well-designed metric is important for the community, especially for the evaluation of latent embedding space.

**Weaknesses:**

I cannot find significant Weaknesses in this paper.

**Questions:**

Would it be possible to make the BSS a training loss to guide and supervise vision encoder embeddings to align with the strong LLM embeddings? Will BSS have additional advantages over contrastive learning loss, such as smaller batch size requirements?

---

### Official Review · Reviewer_E1cX · 2024-11-01

**Soundness:** 3
**Presentation:** 3
**Contribution:** 3
**Rating:** 6
**Confidence:** 2

**Summary:**

This paper introduces a novel semantic metric called Boltzmann Semantic Score (BSS), which is inspired by state space modeling, to evaluate the semantic capability of large vision models (LVMs) in medical image processing. The authors demonstrate the effectiveness of this metric through experiments, revealing that LVMs exhibit low semantic capabilities. Additionally, BSS shows a strong correlation with the performance of LVMs on two clinical tasks: information retrieval and survival prediction.

**Strengths:**

1.The paper is well-structured and clearly presented, which significantly improves its readability.
2.The introduction of the Boltzmann Semantic Score (BSS) is an innovative approach inspired by state space modeling, providing a fresh perspective on evaluating the semantic capabilities of LVMs in medical image processing.
3.The experiments demonstrate significant correlations between BSS and performance on the clinical tasks of information retrieval and survival prediction. Additionally, the experiments show LLMs' capabilities in these two key tasks and provide a quantitative comparison of LLM consistency. This consistency further supports BSS as an effective metric for evaluating the semantic capabilities of LVMs.

**Weaknesses:**

1.The computational complexity of BSS may be high in practical applications, particularly when applied to large-scale datasets.
2.While the experiments show strong performance of BSS in the information retrieval task, its correlation with survival prediction is weaker. This may indicate that BSS lacks robustness across different types of tasks, especially in more complex medical applications. Therefore, its effectiveness as a general semantic metric remains to be further validated.
3.The experiments focus on the tasks of information retrieval and survival prediction, but these tasks may differ in nature from other potential tasks. The consistency of LLMs and the effectiveness of BSS in other semantic tasks require further experimental validation across a broader range of tasks.
4.The paper focuses on evaluating the semantic capabilities of existing LVMs, but it lacks concrete suggestions on how to improve their semantic performance. Although the limitations of LVMs are highlighted, there is little discussion on how to optimize or modify their architectures to overcome these shortcomings.

**Questions:**

1.Could the authors suggest ways to optimize BSS for large-scale datasets, or clarify if any tests on smaller subsets were conducted for comparative analysis?
2. Since BSS performs better on information retrieval than survival prediction, could the authors elaborate on the reasons for this difference? Is there evidence BSS might generalize to other medical tasks?
3. The paper notes limitations in LVMs' semantic capabilities. Do the authors have ideas on potential architectural or training adjustments that might address these limitations?

---

### Official Review · Reviewer_1Uuo · 2024-11-02

**Soundness:** 3
**Presentation:** 2
**Contribution:** 2
**Rating:** 6
**Confidence:** 3

**Summary:**

The paper proposes the Boltzmann Semantic Score (BSS) as a novel metric to evaluate the semantic performance of latent visual models (LVMs) by leveraging large language models (LLMs). The idea behind using BSS is to quantify how well the visual representations align with text expert annotations. The authors show that BSS could be used as a measure of semantic similarity for LVMs. This paper include applications to pathology reports and whole slide images from  The Cancer Genome Atlas (TCGA), a large publicly available cancer genome dataset. Evaluation on various tasks such as information retrieval and survival prediction is included. This paper suggests high correlations for certain cancer between BSS and performance in both survival prediction and information retrieval.

**Strengths:**

•	Boltzmann Semantic Score is a novel approach to evaluate the semantic perspective of LVMs.
•	The work leverages a large dataset (TCGA), and experiments are performed on several benchmarks.
•	The work provides interesting insights on the model performance using BSS based on observed results.
•	High correlation between BSS and two downstream tasks i.e information retrieval and survival prediction, highlighting the significance of the results.
•	Interesting experiments on clinical tasks showing correspondence between LLMs and patient survival.

**Weaknesses:**

•	The mathematics for the explanation of the Boltzmann Score and its application is rather heavy. A more concise and clearer explanation would enable to understand better the intuition behind the usefulness of BSS as an evaluation metric for LVMs.
•	The authors could develop more on the clinical implication and real-word use of BSS in decision-making.
•	Some insights are provided to explain the differences between LVMs and LLMs performance, but the paper could investigate more thoroughly those differences and inherent variations.
•	A discussion of the limitations of this work in terms of generalization under different contexts is lacking.

**Questions:**

•	Did you visualize the semantic similarity and qualitatively assess the use of BSS as evaluation metric?
•	How reliable is Boltzmann Semantic Score ?
•	What preprocessing was applied to the medical reports?
•	Could you explain the differences observed in Table 3 a) for the two-sided Pearon's Correlation Test?
•	What is the effect of bias originating from the datasets ?

---

### Public Comment · ~Ali_Khajegili_Mirabadi1 · 2026-04-29
**Funding Acknowledgements**

This work was supported in part by a Marathon of Hope award through the Terry Fox Research Institute (TFRI), grant number 3262-03.

---

### Meta-Review · Area_Chair_2WTX · 2024-12-21

**Metareview:**

The paper introduces the Boltzmann Semantic Score (BSS), a novel metric for evaluating the semantic capabilities of large vision models (LVMs) using large language models (LLMs) and medical reports. The authors demonstrate BSS's effectiveness with extensive experiments, showing strong correlations with performance on tasks like information retrieval and survival prediction. Some concerns were raised about BSS's scalability to large datasets and its robustness across different tasks. Despite these points, the paper is seen as highly original and important for the medical imaging community, with one reviewer suggesting the possibility of incorporating BSS as a training loss for further optimization. Overall, the paper is recommended for acceptance, with minor improvements suggested.

**Additional Comments On Reviewer Discussion:**

Before the discussion stage, reviewers noted that the explanation of BSS's mathematical foundations could be clearer, and that the paper lacked a discussion of its limitations and practical applications. However, these concerns appear to have been effectively addressed during the discussion.

---

### Decision · Program_Chairs · 2025-01-22

Accept (Poster)